# A *Phytophthora* receptor-like kinase regulates oospore development and can activate pattern-triggered plant immunity

Yong Pei[1], Peiyun Ji[1], Jierui Si[1], Hanqing Zhao[1], Sicong Zhang[1], Ruofei Xu[1], Huijun Qiao[1], Weiwei Duan[1], Danyu Shen[1], Zhiyuan Yin [1] ✉ & Daolong Dou [1,2] ✉

Plant cell-surface leucine-rich repeat receptor-like kinases (LRR-RLKs) and receptor-like proteins (LRR-RLPs) form dynamic complexes to receive a variety of extracellular signals. LRR-RLKs are also widespread in oomycete pathogens, whereas it remains enigmatic whether plant and oomycete LRR-RLKs could mediate cell-to-cell communications between pathogen and host. Here, we report that an LRR-RLK from the soybean root and stem rot pathogen *Phytophthora sojae*, PsRLK6, can activate typical pattern-triggered immunity in host soybean and nonhost tomato and *Nicotiana benthamiana* plants. PsRLK6 homologs are conserved in oomycetes and also exhibit immunity-inducing activity. A small region (LRR5-6) in the extracellular domain of PsRLK6 is sufficient to activate BAK1- and SOBIR1-dependent immune responses, suggesting that PsRLK6 is likely recognized by a plant LRR-RLP. Moreover, *PsRLK6* is shown to be up-regulated during oospore maturation and essential for the oospore development of *P. sojae*. Our data provide a novel type of microbe-associated molecular pattern that functions in the sexual reproduction of oomycete, and a scenario in which a pathogen LRR-RLK could be sensed by a plant LRR-RLP to mount plant immunity.

Plant transmembrane receptor-like kinases (RLKs) and receptor-like proteins (RLPs) play important roles in growth, development, and defense responses[1,2]. The various extracellular domains (ECDs) can sense microbe-associated molecular patterns (MAMPs) during interaction to mount pattern-triggered immunity (PTI). Microbe-associated molecular patterns (MAMPs) are generally considered to be highly conserved throughout classes of microbes and contribute to microbial fitness[3]. The majority of currently characterized MAMPs are molecules secreted by microbes or cell components of microbes that are released by plants[4,5]. For example, plant enzymes target the bacterial flagella and fungal cell wall to release flagellin[6] and chitin[7] respectively, while Nep1-like proteins[8] and glycoside hydrolases[9] are virulence factors that are also recognized as MAMPs by plants. MAMP-receptor pairs are useful for guiding plant-resistance breeding. Recently, transgenic wheat expressing the *Nicotiana benthamiana* receptor RXEG1

exhibited enhanced resistance to *Fusarium graminearum* by recognizing glycoside hydrolase 12 MAMPs[10]. There are hundreds of RLKs and RLPs on the plant cell surface, but only dozens of MAMPs were found up till now[2].

Oomycetes resemble filamentous fungi in morphology but belong to Stramenopiles. *Phytophthora* is an oomycete genus that includes ~200 destructive plant pathogens, causing a variety of agricultural losses worldwide as well as environmental damage in natural ecosystems[11]. For example, potato late blight caused by *P. infestans*, which was one of the causes of the Irish famine of the 19th century and which still threatens the safe production of potatoes and tomatoes[12]. *P. sojae* causes soybean root and stem rot and has been developed as a model species for the study of oomycete development and pathogenesis owing to the availability of abundant genomic data and well-established gene editing technique[11,13,14]. The leucine-rich repeat (LRR)

[1]College of Plant Protection, Nanjing Agricultural University, Nanjing 210095, China. [2]Academy for Advanced Interdisciplinary Studies, Nanjing Agricultural University, Nanjing 210095, China. ✉e-mail: zyin@njau.edu.cn; ddou@njau.edu.cn

RLKs form the largest family of plant cell-surface receptors. Interestingly, LRR-RLKs with a similar domain structure are also widespread in oomycetes although they are evolutionarily distant to plants[15,16]. We previously reported that PcLRR-RK1 regulates the growth, development, and virulence of *P. capsici*[17] and several *P. sojae* LRR-RLKs play pivotal roles in zoospore production and virulence[16].

LRR-containing proteins may interact with each other to regulate different biological processes. For MAMPs (microbe-associated molecular patterns) recognition, interactions between ECDs of LRR-RLKs are a prerequisite for ligand signal activation[18]. In *Arabidopsis*, an extracellular network consisting of 567 interactions among ECDs of 225 LRR-RLKs was identified[19]. A similar extracellular interactome of LRR proteins was revealed in *Drosophila melanogaster*[20]. Similar to plants and metazoans, *P. sojae* also has a complicated interaction network of LRR-RLKs[16]. Importantly, it is worth to notice that receptors with LRR ECDs between adjacent cells can also establish contacts and mediate cell-to-cell communication in metazoans[21].

Given that both *P. sojae* and the host plant possess many LRR-RLKs that are essential for virulence or resistance, we hypothesized that they could mediate cell-to-cell communications during interaction. To test it, we systematically expressed 24 *P. sojae* LRR-RLKs and found that PsRLK6 could elicit immune responses in *Nicotiana benthamiana*. PsRLK6 is conserved and widespread across Peronosporales, whose homologs exhibited immune-inducing activity in the plant as well. Moreover, the immunity activated by PsRLK6 required two co-receptors BAK1 and SOBIR1, suggesting that an LRR-RLP binds PsRLK6 to mount plant immunity. Our data provide a novel type of microbe-associated molecular patterns (MAMP) that is a cell surface-localized RLK and a scenario showing plants and pathogens communicate with each other via cell-surface receptors.

## Results

### *P. sojae* PsRLK6 can activate typical pattern-triggered immunity (PTI) in *Nicotiana benthamiana*

In our previous study, a total of 24 *LRR-RLK* genes (*PsRLKs*) were identified in the *P. sojae* genome[16]. We hypothesized that some PsRLKs could interact with plant receptors to induce plant immunity during infection. Thus, extracellular domains (ECDs) of all 24 PsRLKs were cloned and then transiently expressed in *N. benthamiana* by agroinfiltration. The DAB (3,3-diaminobenzidine) staining assay revealed that PsRLK6ECD and PsRLK7ECD could induce reactive oxygen species (ROS) accumulation in *N. benthamiana*, while other PsRLKs and the negative control GFP could not (Fig. 1a). To further examine the effect of PsRLK6ECD and PsRLK7ECD on plant resistance to pathogen, *N. benthamiana* leaves that expressed PsRLK6ECD, PsRLK7ECD, or GFP for 24 h were inoculated with *P. capsici*. Leaf regions expressing PsRLK6ECD or PsRLK7ECD showed significantly smaller lesion areas and less *P. capsici* biomass, compared to that expressing GFP (Supplementary Fig. 1a, b). Western blot analysis showed that GFP, PsRLK6ECD, and PsRLK7ECD were successfully expressed at the expected size in *N. benthamiana* (Supplementary Figs. 1c, d). These results suggest that *P. sojae* PsRLKs could enhance plant resistance in *N. benthamiana*.

To further confirm that PsRLK6ECD and PsRLK7ECD could induce immune responses in *N. benthamiana*, we initially expressed and purified His-tagged PsRLK6ECD and PsRLK7ECD from *Pichia pastoris* (Supplementary Fig. 2). The recombinant proteins were used to test for a series of PTI (pattern-triggered immunity) responses, including ROS burst, MAPK activation, and up-regulation of two PTI marker genes (*PR1a* and *CYP71D20*). As shown in Fig. 1b, 1 μM PsRLK6ECD protein treatment increased ROS levels after about 5 min upon elicitation, reaching the maximum levels at 30 min. However, PsRLK7ECD protein failed to induce detectable ROS production even at 60 min, similar to GFP protein (Fig. 1b). Therefore, PsRLK6ECD was selected for further studies. To test whether PsRLK6ECD-induced ROS burst is concentration-dependent, different concentrations of PsRLK6ECD proteins

ranging from 0.01 μM to 5 μM were examined. The result showed that the total ROS production of 60 min increased with higher concentrations (Fig. 1c).

MAPK activation is another early PTI (pattern-triggered immunity) signaling event. As shown in Fig. 1d, PsRLK6ECD protein treatment triggered MAPK activation at 5, 10, and 15 min in *N. benthamiana* as detected by immunoblot with the α-pTEpY antibody. We also observed that the expressions of PTI marker gene *PR1a* and *CYP71D20* were greatly activated by PsRLK6ECD protein for 6 h (Fig. 1e, f). Finally, to examine the effect of PsRLK6ECD protein on plant resistance to pathogen, *N. benthamiana* leaves were pre-treated with 1 μM PsRLK6ECD or GFP proteins, respectively, and were inoculated with *P. capsici* 24 h later. Consistent with the above results, pre-treatment with PsRLK6ECD protein resulted in significantly smaller lesions and less *P. capsici* biomass compared to GFP (Fig. 1g–i). Collectively, these findings show that PsRLK6ECD can activate PTI (pattern-triggered immunity) responses in *N. benthamiana*.

### BAK1 and SOBIR1 are required for PsRLK6-induced immune responses in *N. benthamiana*

LRR-RLKs BAK1 and SOBIR1 are two co-receptors of most LRR-RLK and LRR-RLP receptors for signaling activation[4]. To determine whether PsRLK6ECD-induced immune responses are BAK1- and SOBIR1-dependent in *N. benthamiana*, we used the virus-induced gene silencing (VIGS) construct to silence *BAK1*. The *sobir1* mutant was generated by CRISPR/Cas9 previously[22]. PsRLK6ECD-induced ROS production, activation of MAPK and the PTI marker gene *CYP71D20* were measured in *BAK1*-silenced and *sobir1* plants. The results showed that the ROS production was remarkably compromised in *BAK1*-silenced plants and *sobir1* mutants compared to *GFP*-silenced and wild-type plants (Fig. 2a–d). The relative expressions of *CYP71D20* induced by PsRLK6ECD protein were reduced by about 12-fold in *BAK1*-silenced plants and 40-fold in *sobir1* mutants, respectively (Fig. 2e, f). Similarly, the MAPK activation was also remarkably compromised in *BAK1*-silenced plants and *sobir1* mutants (Fig. 2g, h). We next evaluated whether BAK1 and SOBIR1 are also required for PsRLK6ECD-induced resistance to *P. capsici*. Compared with leaves treated with TRV2:*GFP*, *BAK1*-silenced leaves are more susceptible to *P. capsici* and PsRLK6ECD-induced resistance was blocked through monitoring lesion areas and relative pathogen biomass in *BAK1*-silenced plants (Supplementary Fig. 3a–d). Similar results were observed in the *sobir1* mutant, which showed increased disease severity and comprised plant resistance induced by PsRLK6ECD (Supplementary Fig. 3e–g). Collectively, these results showed that BAK1 and SOBIR1 are both required for PsRLK6-triggered immunity in *N. benthamiana*.

### PsRLK6 can trigger immune responses in soybean and tomato

We also tested the elicitor activity of PsRLK6ECD in the cognate host soybean (*Glycine max*) along with two nonhost plants tomato and *Arabidopsis thaliana*. The results showed that 1 μM PsRLK6ECD protein could trigger ROS burst in soybean and tomato, but not in *A. thaliana* (Fig. 3a–c). Similarly, the marker gene *PR1a* was up-regulated in soybean and tomato but not in *A. thaliana* after being pre-treated for 6 h with PsRLK6ECD compared to GFP protein (Fig. 3d–f). To further evaluate the contribution of PsRLK6ECD to disease resistance in soybean, the etiolated soybean hypocotyls were placed into 50 nM PsRLK6ECD and GFP protein solution respectively. Twelve hours later, soybean hypocotyls were taken out from the protein solution and inoculated with a piece of mycelium of *P. sojae*. Compared to the GFP control, pre-treatment with PsRLK6ECD resulted in lighter symptoms in plants (Fig. 3g). Similarly, 50 nM GFP and PsRLK6ECD protein were used to treat tomato and *A. thaliana*, and PsRLK6ECD could induce resistance to *P. capsici* in tomato (Fig. 3h) but not in *A. thaliana* (Fig. 3i). These results suggest that PsRLK6ECD could induce immune responses in multiple plants including the host plant soybean.

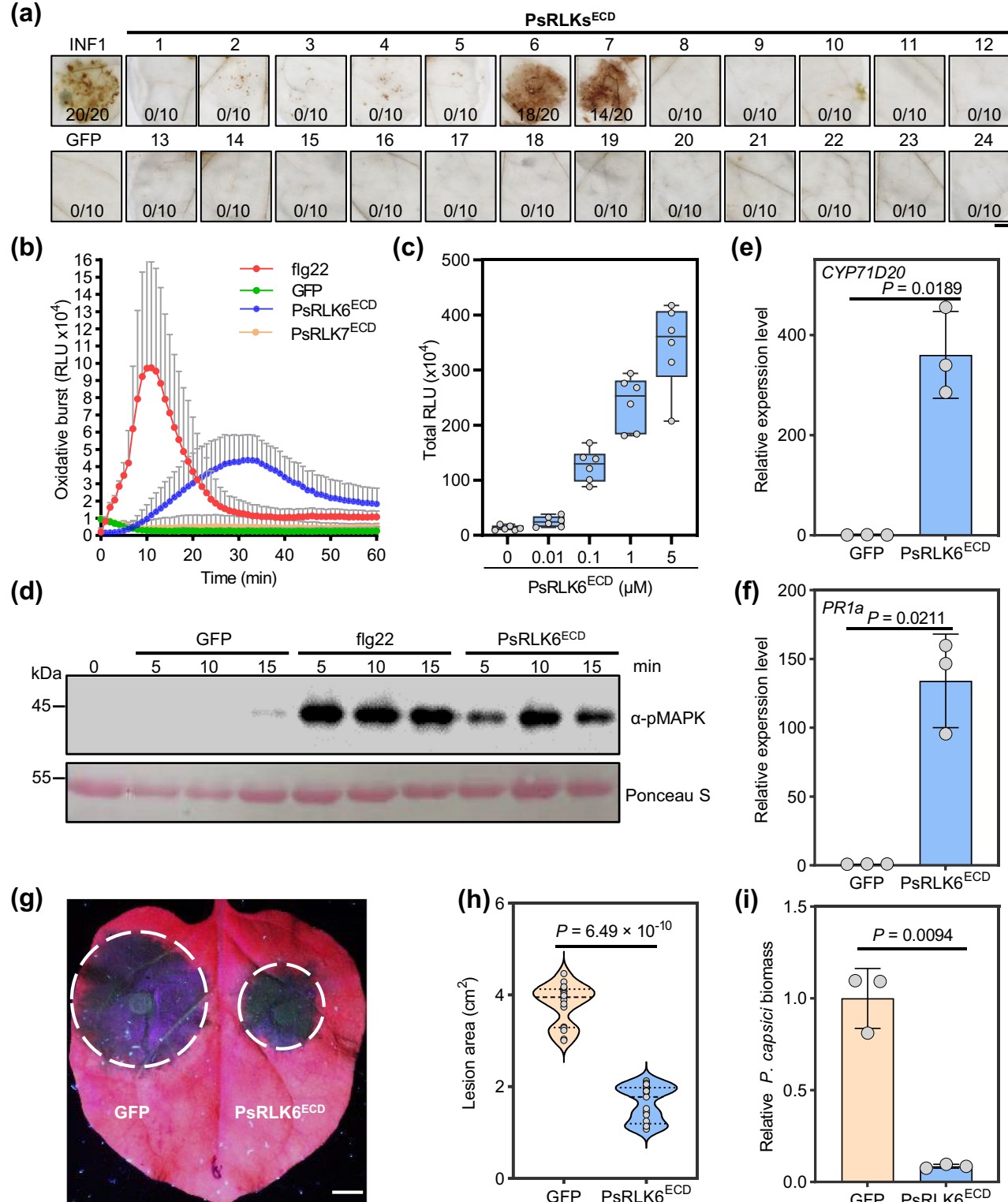

## A small region (LRR5-6) of PsRLK6 is sufficient to elicit immune responses in plants

To identify the critical area of the PsRLK6 for elicitor activity, we firstly predicted the 3D structure of PsRLK6ECD by AlphaFold 2[23]. Total of five generated models were provided, and the pLDDT coloring was displayed in Fig. 4a. The best predicted structure of PsRLK6ECD was presented, wherein the ECD forms a single continuous structure in an arc shaped conformation (Fig. 4a). The inner face of the arc forms a concave surface, the majority of which contain an extended parallel

β-sheet. The outer face forms a convex side mostly consisting of various secondary structures such as α-helices, loops and turns. According to the predicted structure, PsRLK6ECD contains an N-terminal signal peptide domain, an LRR capping domain, seven atypical LRR domains, and an LRR C-terminal domain (Fig. 4b). To test whether the elicitor activity of PsRLK6ECD is heat stable, we treated the PsRLK6ECD protein by boiling at 100 °C for 10 min. The results showed that boiled PsRLK6ECD protein still could induce ROS accumulation (Fig. 4c) and disease resistance to *P. capsici* in *N. benthamiana* (Supplementary Fig. 4). So,

**Fig. 1 | PsRLK6 can induce PTI responses in *N. benthamiana*. a** PsRLK6/7 induced ROS accumulation in *N. benthamiana* leaves. The 24 different PsRLKs^ECD, positive control INF1 and negative control GFP were transiently expressed in *N. benthamiana* for 24 h and then detected by DAB staining (*n* = 10 or 20 samples). **b** The recombinant PsRLK6^ECD protein induced ROS burst in *N. benthamiana*. Leaf discs from plants were assayed for ROS production by measuring the relative light units (RLU) with a luminometer upon GFP or PsRLK6^ECD treatment for the indicated time points. Data are shown as mean ± SD (*n* = 8). **c** PsRLK6^ECD induced ROS burst in a concentration-dependent manner. Accumulation of relative light units (RLU) in 60 min was calculated for *N. benthamiana* treated with different indicated concentrations of PsRLK6^ECD (*n* = 6 biologically independent samples). **d** PsRLK6^ECD-induced MAPK activation in *N. benthamiana*. The plants were infiltrated with PsRLK6^ECD (1 μM), flg22 and GFP protein for 5, 10, and 15 min. MAPK activation was analyzed by immunoblot with the α-pMAPK antibody. Protein loading is shown

using Ponceau S. **e**, **f** PsRLK6^ECD induced the activation of PTI marker genes (*n* = 3 biologically independent experiments). **g** PsRLK6^ECD enhanced plant resistance to *P. capsici*. Half leaves of plants were infiltrated with 1 μM PsRLK6^ECD or GFP proteins for 24 h before inoculation with *P. capsici*. Photos of the symptom of *N. benthamiana* leaves were taken 36 h post inoculation. **h** Lesion areas on *N. benthamiana* leaves caused by *P. capsici* (*n* = 17 biologically independent samples). **i** Quantification of *P. capsici* biomass by qRT-PCR analysis to measure the ratios of *P. capsici* to *N. benthamiana* DNA. Values are presented as mean ± SD (*n* = 3 biologically independent experiments). In (**c**) and (**h**), the data are shown as box plots and violin plots, respectively. The center line, edges and whiskers indicate the median, lower and upper quartiles and the minimum and maximum, respectively. The statistical analyses were performed with two-tailed Student's *t* test. Scale bars, 1 cm. The experiments (**a**, **b**, **c**, **d**, **g**, **h**) were repeated at least three times. Source data are provided as a Source Data file.

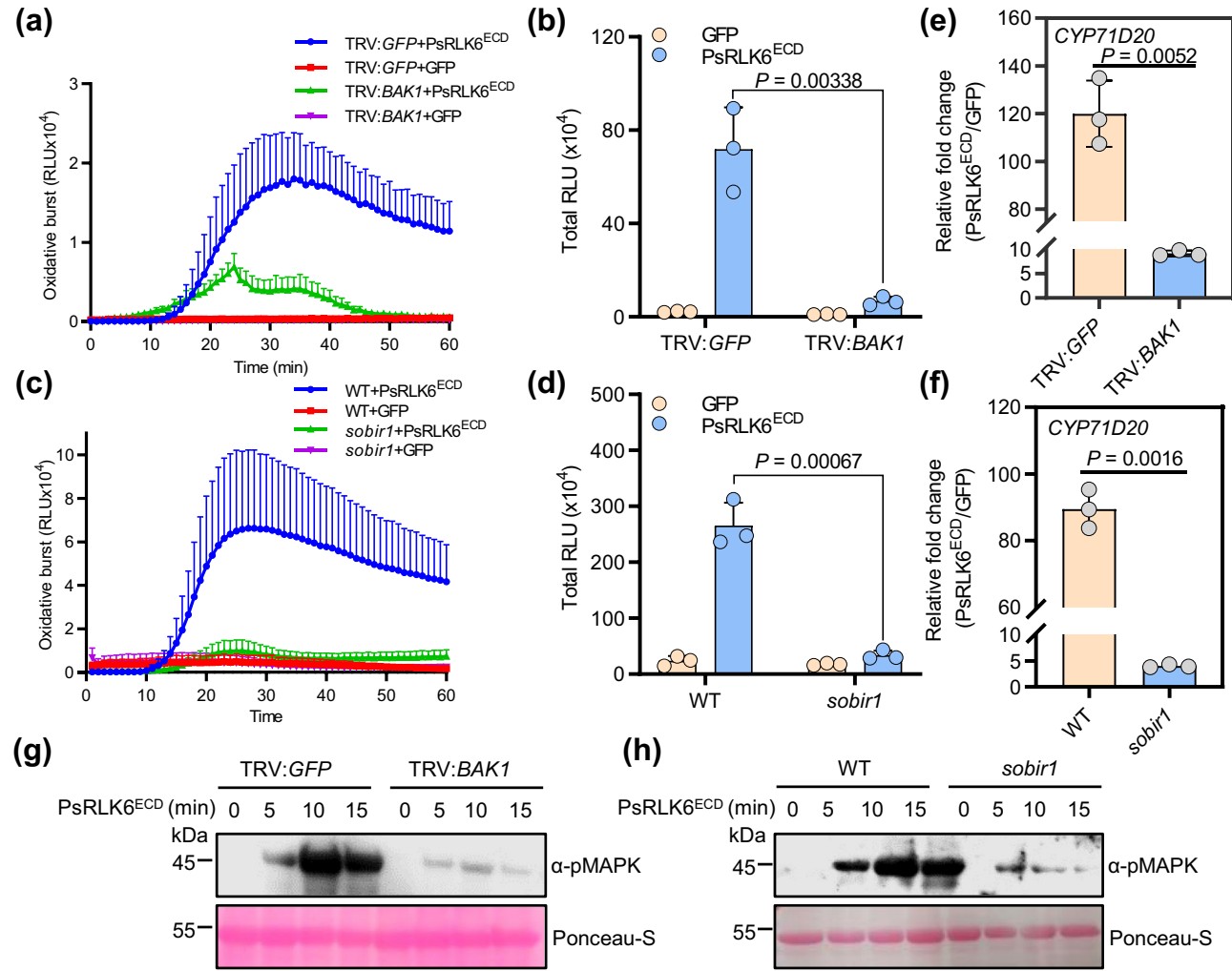

**Fig. 2 | Immune responses triggered by PsRLK6 require coreceptors BAK1 and SOBIR1. a** PsRLK6^ECD-induced ROS burst is dependent on BAK1 in *N. benthamiana*. The 2-week-old *N. benthamiana* plants were subjected to virus-induced gene silencing (VIGS) by inoculation of tobacco rattle virus (TRV) constructs (TRV2:*GFP* and TRV2:*BAK1*). Three weeks after agroinfiltration, ROS production induced by GFP or PsRLK6^ECD proteins for the indicated time points were measured. Data are shown as mean ± SEM (*n* = 8 biologically independent samples). **b** Total RLU-induced by GFP or PsRLK6^ECD for first 60 min in TRV2:*GFP*- or TRV2:*BAK1*-treated *N. benthamiana* plants. Data are shown as mean ± SD (*n* = 3 biologically independent experiments). **c** PsRLK6^ECD-induced ROS burst is dependent on SOBIR1 in

*N. benthamiana*. Data are shown as mean ± SEM (*n* = 8 biologically independent samples). **d** Total RLU-induced by GFP or PsRLK6^ECD in wild-type or *sobir1* plants. Data are shown as mean ± SD (*n* = 3 biologically independent experiments). **e**, **f** PsRLK6^ECD-induced up-regulation of the PTI marker gene *CYP71D20* requires BAK1 and SOBIR1 in *N. benthamiana*. The data are presented as mean ± SD (*n* = 3 biologically independent experiments). **g**, **h** PsRLK6^ECD induces the MAPK activation in a BAK1- and SOBIR1-dependent manner in *N. benthamiana*. The experiments (**a**, **c**, **g**, **h**) were repeated three times with similar results. *P* values were derived by two-tailed Student's *t*-test. Scale bars, 1 cm. Source data are provided as a Source Data file.

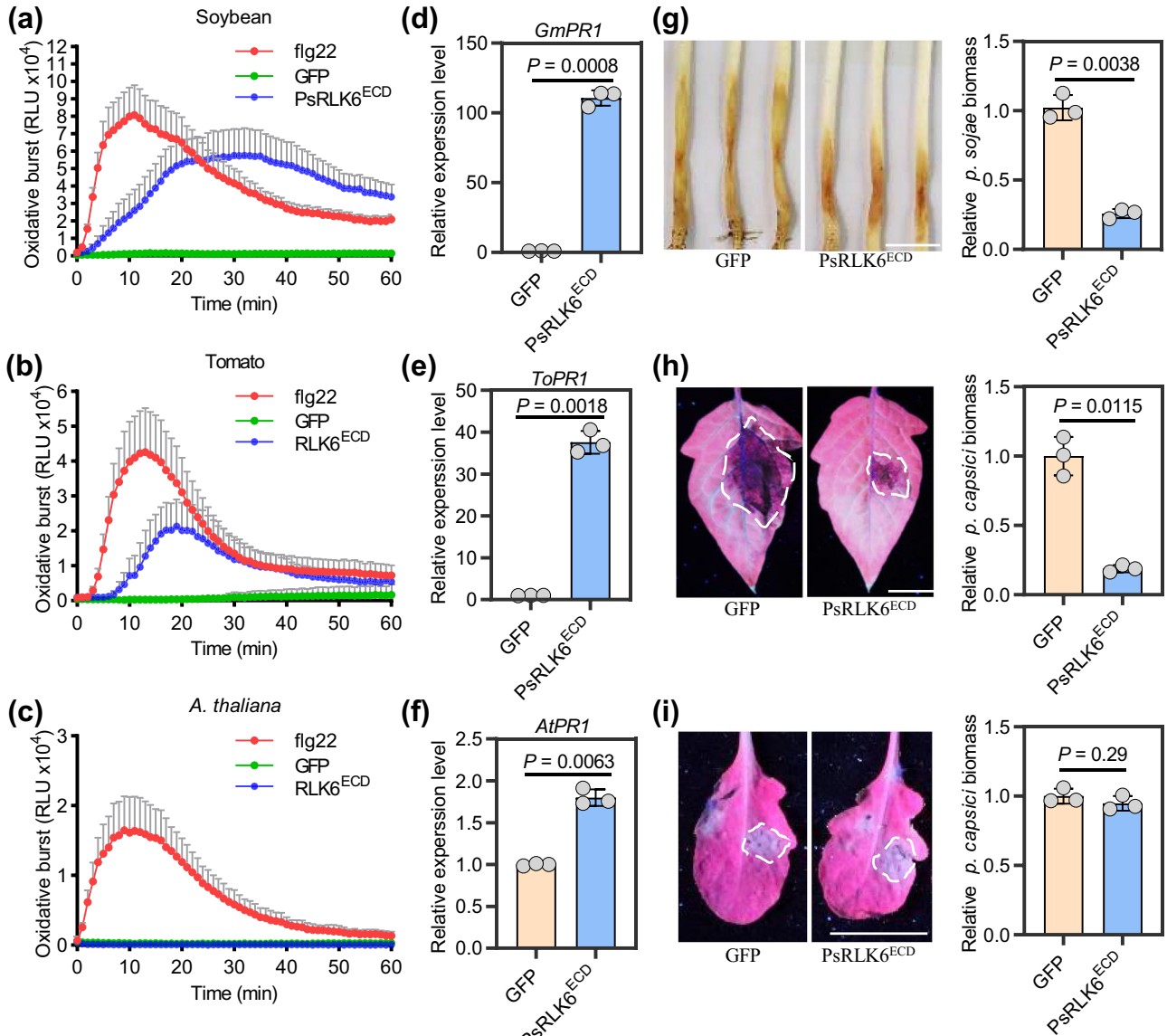

**Fig. 3 | PsRLK6 can induce immune responses in soybean and tomato.**
PsRLK6$^{ECD}$ induced ROS burst in soybean (**a**) and tomato (**b**) but not in *A. thaliana* (**c**). Leaf discs from different plants were assayed for ROS production by measuring the relative light units (RLU) with a luminometer upon GFP protein (negative control), flg22 peptide (positive control) or PsRLK6$^{ECD}$ treatment for the indicated time points. Data are shown as means ± SD (*n* = 8 biologically independent samples). **d**–**f** The relative expression of *PR1a* in PsRLK6$^{ECD}$ infiltrated plant leaves mentioned above. GFP was used as the control for PsRLK6$^{ECD}$ protein treatment. Soybean *Actin*, tomato *Actin* and *Arabidopsis ACT2* genes were used as references, respectively (*n* = 3 biologically independent experiments). **g** PsRLK6$^{ECD}$ induced disease resistance to *P. sojae*. Etiolated soybean hypocotyls were soaked with 50 nM

PsRLK6$^{ECD}$ or GFP proteins for 12 h. Lesion lengths were assessed 48 hpi. Relative biomass of *P. sojae* in the infected etiolated soybean hypocotyls was measured by qRT-PCR and was normalized to that treated with GFP (*n* = 3 biologically independent experiments). **h** Induction of disease resistance to *P. capsici* induced by spray pretreatment of tomato leaves with 50 nM PsRLK6$^{ECD}$ for 24 h compared to GFP (*n* = 3 biologically independent experiments). **i** Spray pretreatment of PsRLK6$^{ECD}$ in *A. thaliana* could not induce disease resistance to *P. capsici* (*n* = 3 biologically independent experiments). The experiments (**a**, **b**, **c**) were performed three times with similar results. *P* values were derived by two-tailed Student's *t*-test. Scale bars, 1 cm. Source data are provided as a Source Data file.

we hypothesized an epitope might be recognized by plants to elicit immune responses. To map the PsRLK6$^{ECD}$ epitope that is responsible for the elicitor activity, we generated truncated mutant variants of PsRLK6$^{ECD}$ that carried an intact signal peptide and tested their ability to test the ROS accumulation in *N. benthamiana* by agroinfiltration (Fig. 4b). As shown in Fig. 4b, the region containing residues 237–281 (M7), corresponding to LRR5-6, fully triggered ROS accumulation in *N. benthamiana* leaves (Fig. 4b). Furthermore, pathogen inoculation assays confirmed that expression of LRR5-6 in *N. benthamiana* was sufficient to enhance plant resistance to *P. capsici* (Fig. 4b). These results indicate that the LRR5-6 of PsRLK6$^{ECD}$ is the key region for

elicitor function. To further verify whether the immunogenic epitope LRR5-6 could trigger plant immunity, the peptide was chemically synthesized. However, we failed to obtain the final peptide after trying three times. Furthermore, we expressed and purified the LRR5-6 region using *E. coli* to investigate whether it could elicit immune responses in different plants. The results demonstrated that LRR5-6 could induce ROS burst in *N. benthamiana* (Fig. 4d), and subsequent experiments confirmed its ability to trigger ROS burst in soybean (Fig. 4e) and tomato (Fig. 4f). In conclusion, these results demonstrate that two LRR motif of PsRLK6$^{ECD}$ is sufficient to elicit immune response in multiple plants.

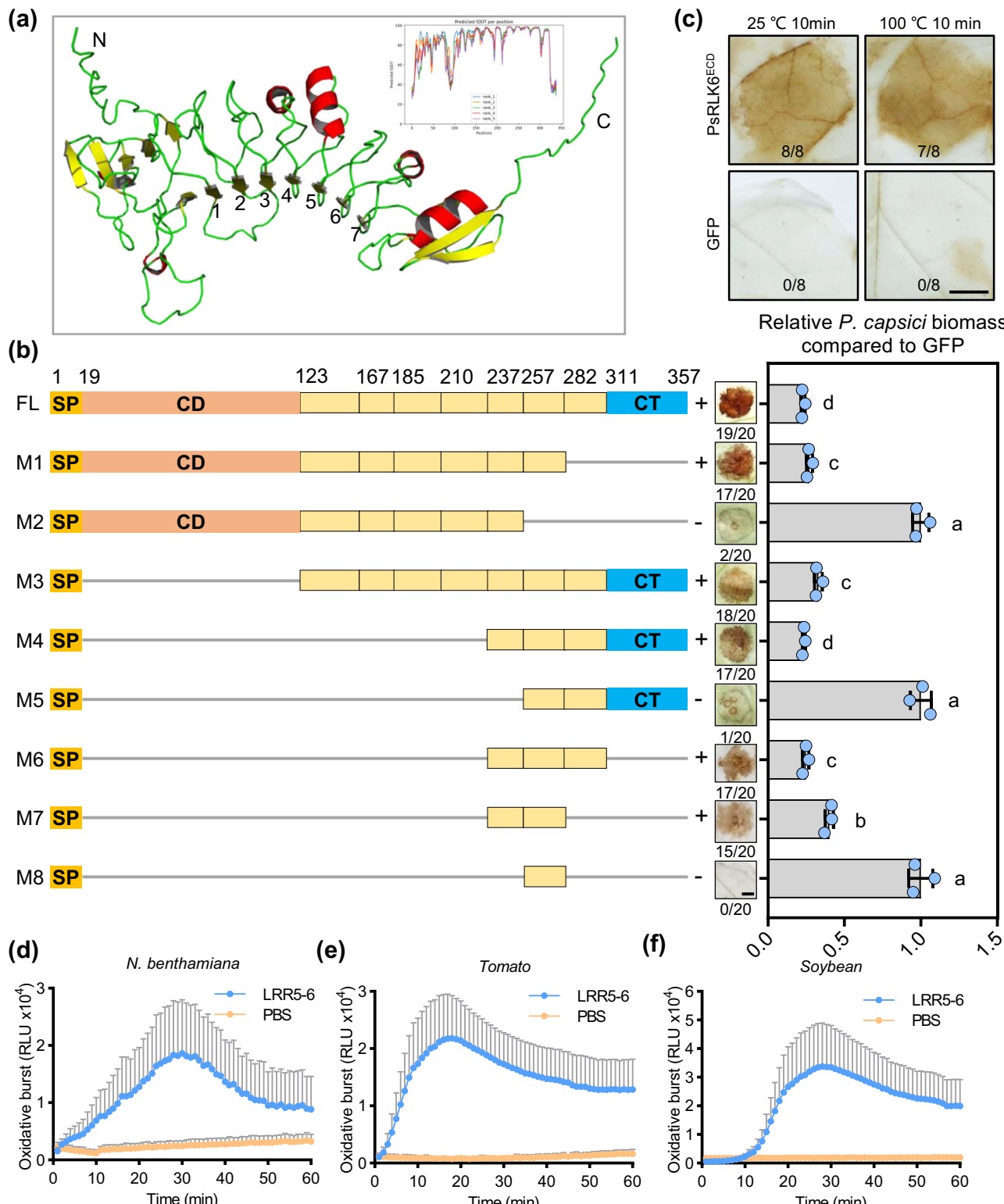

**The elicitor function of RLK6 could be conserved in oomycetes**

Since LRR-RLKs are widespread in oomycetes and plants, but not in diatoms, fungi, or metazoans[16], to investigate the phylogenetic distribution of PsRLK6 homologs, we queried the genomes of twenty-six oomycete, three fungal, two bacterial, and two plant species. A total of 53 homologous protein sequences were obtained in different oomycete species, while no homologs were found in others (Fig. 5a, b, Supplementary Data 1). Interestingly, PsRLK6 homologs were only found in the Peronosporales but not in Saprolegniales (Fig. 5a, b). Most

oomycetes contain no more than three PsRLK6 homologs, while *Phytophthora fragariae* and *Phytophthora cactorum* have much more homologs than other species (Fig. 5b). Additionally, the Consurf server[24] was used to analyze the conservation of the structure of obtained 53 homologous protein sequences with PsRLK6^ECD. The results showed that they share a high similarity at the protein level (Supplementary Fig. 5a). The enlarged image of the key region for elicitor function LRR5-6 also showed a high similarity (Supplementary Fig. 5b).

**Fig. 4 | Two LRR motifs of PsRLK6 are sufficient to induce immune responses in plants. a** Structure of the extracellular domain of PsRLK6. The structure of PsRLK6[ECD] (residues 19–357) is represented as a cartoon representation. The α-helix is colored in red, β-strand in yellow, and loop in cyan. The LRR motif is numbered in the N-terminal β-strand of the repeat from 1–7. **b** Regions of PsRLK6[ECD] examined for ROS-inducing activity by *Agrobacterium*-mediated transient expression. PsRLK6[ECD] contains an N-terminal signal peptide domain (SP, residues 1–19), an LRR capping domain (CD, 20–122), a leucine-rich repeat domain (LRR, 123–310) which harbors seven LRRs, and an LRR C-terminal domain (CT, 311–357). The corresponding ROS accumulation (*n* = 20 biologically independent samples) and disease resistance to *P. capsici* (*n* = 3 biologically independent experiments) were shown in the right panel. Bars indicate mean ± SD. The statistical analyses were performed with two-way and different letters show significant difference. **c** ROS accumulation induced by infiltration of heat-treated PsRLK6[ECD] in *N. benthamiana* leaves. The PsRLK6[ECD] protein was treated at 100 °C for 10 min. Ratios indicate the proportion of infiltrated sites that developed the ROS accumulation phenotype (*n* = 8 biologically independent samples). LRR5-6 region of PsRLK6[ECD] induced ROS burst in *N. benthamiana* (**d**), soybean (**e**) and tomato (**f**). Leaf discs from *N. benthamiana*, soybean and tomato were assayed for ROS production by measuring the relative light units (RLU) with a luminometer upon PBS buffer (negative control) or LRR5-6 treatment for the indicated time points (*n* = 8 biologically independent samples). The experiment repeated three times with similar results. Data are shown as means ± SEM. Scale bars, 1 cm. Source data are provided as a Source Data file.

To investigate whether the PsRLK6 homologs from diverse *Phytophthora* and *Pythium* species could induce immune responses, we cloned the ECDs of six homologous genes from *Phytophthora infestans*, *Phytophthora capsici*, *Phytophthora parasitica*, *Phytophthora nicotianae*, *Phytophthora cactorum*, and *Pythium oligandrum* (Supplementary Fig. 5c). These homologs were transiently expressed in *N. benthamiana* to test for their ability to induce ROS accumulation. Similar to PsRLK6[ECD], DAB staining assays showed that all six PsRLK6[ECD] homologs could induce ROS accumulation in *N. benthamiana* (Fig. 5c). Western blot analysis confirmed that all the proteins were successfully expressed at the expected size in *N. benthamiana* (Fig. 5d). Moreover, expression of these six homologs in *N. benthamiana* all resulted in remarkably reduced *P. capsici* biomass compared to the GFP control (Fig. 5e). These results indicated that conserved PsRLK6 homologs from diverse oomycetes could be recognized by *N. benthamiana* to mount plant immunity.

## PsRLK6 can trigger immune responses during *P. sojae*-soybean interaction

MAMPs often play important roles in the growth, development, and virulence of microbes. However, we previously reported that the *PsRLK6* knockout mutants have no obvious phenotypes on growth, zoospore development, and virulence[16]. A transcriptome data showed that *PsRLK6* was up-regulated during infection stages[25], suggesting its potential role during interaction. Therefore, we over-expressed *PsRLK6-GFP* by PEG-mediated transformation[26]. Immunoblotting analysis using the α-GFP antibody confirmed that the protein was expressed in the mycelia of the OT24 and OT36 lines (Supplementary Fig. 6a). All transformants showed normal filamentous growth compared with WT (Supplementary Fig. 6b, c). To assay the virulence of the transformants, zoospores were inoculated onto etiolated seedlings of soybean. The knockout mutant ΔPsRLK6 showed unchanged lesion length as reported previously[16], while overexpression of *PsRLK6-GFP* showed significantly reduced virulence on soybean (Supplementary Fig. 6d). Determination of the *P. sojae* DNA biomass confirmed that virulence of OT24 and OT36 transformants was severely attenuated compared to that of WT and ΔPsRLK6 (Supplementary Fig. 6e, f). These results indicate that overexpression of PsRLK6 could limit *P. sojae* infection, likely by inducing defense responses.

Furthermore, we conducted a series of experiments to investigate whether PsRLK6 could elicit PTI immune responses during *P. sojae*-soybean interaction, including ROS burst, up-regulation of *PR1a* and MAPK activation. At 24 h post-inoculation (hpi), PsRLK6 over-expression line OT24 exhibited a higher level of ROS accumulation compared to both the WT and knockout mutant, which displayed similar levels of ROS staining as the WT after DAB staining (Fig. 6a). Next, the ROS burst in soybean leaves triggered by different strains was quantified, revealing that PsRLK6 knockout mutants produced a slightly reduced amount of ROS compared to the WT, while overexpression lines generated significantly higher levels of ROS (Fig. 6b, c). Similarly, the fold change of *PR1a* expression was lower than that induced by the WT, whereas OT24 overexpression line exhibited an opposite trend (Fig. 6d). Further experiments on MAPK activation of different strains revealed that OT24 exhibited greater MAPK activation compared to WT and ΔPsRLK6 exhibited a little less (Fig. 6e), indicating a potential role of PsRLK6 in inducing MAPK activation of soybean. Taken together, these results show that PsRLK6 can trigger defense responses, though the extent to which it does so during infection of plants with wild-type *Phytophthora* remains to be determined.

## PsRLK6 is required for the oospore development of *P. sojae*

In nature, survival and dispersal of spores are essential for the success of many plant pathogens. The sexual reproduction of *P. sojae* produces thick-walled oospores, which are important inoculum to trigger a new epidemic in the next growing season[27]. Sexual reproduction is particularly relevant for the homothallic *P. sojae*[28]. To further explore the biological role of *PsRLK6* during the sexual cycle of *P. sojae*, we firstly investigated the number and morphology of oospores produced by *PsRLK6* knockout and overexpression transformants. Interestingly, after culturing on LBA (lima bean agar) medium for 14 days, the knockout mutant generated considerably reduced oospores, while overexpression lines produced slightly more oospores than WT (Fig. 7a, b). The oospores produced by ΔPsRLK6 showed more abnormal morphological oospores while overexpression lines showed less (Fig. 7c). We further determined whether *PsRLK6* could up-regulate in oospore development stages. The mycelia grown on the LBA medium for 6, 12, and 20 days produced no oospores, immature oospores and mature oospores, respectively. We found that *PsRLK6* is up-regulated about 3-fold and 12-fold in the stages of immature and mature oospores respectively (Fig. 7d). Furthermore, we investigated oospore number and morphology produced by different strains during infection. Due to phenotype of two overexpression lines grown on LBA medium was consistent, we selected OT24 for further study. Oospores were generated on infected root tissue with mycelia at 48 hpi (top row) and 96 hpi (bottom row) stained with lactophenol-trypan blue. The results showed that oospores produced in infected soybean hypocotyl tissue by *PsRLK6* knockout mutant was less than WT, while *PsRLK6* overexpression transformant (OT24) produced more oospores (Fig. 7e, f). Similar to that grown on the LBA medium, the oospores produced by ΔPsRLK6 during infection showed more abnormal morphological oospores, while overexpression line (OT24) showed less (Fig. 7g). We further determined whether *PsRLK6* could up-regulate in infection stages when oospores could be produced. In the late stages of infection, the oospores became mature gradually, and the expression levels of *PsRLK6* were significantly increased (Fig. 7h). Taken together, these results indicate that PsRLK6 is essential for the oospore development of *P. sojae*.

## Discussion

To overcome pathogen attacks, plants have evolved an excellent system to recognize non-self MAMPs (microbe-associated molecular patterns) through cell-surface receptors, among which LRR-RLKs are the largest family of RLKs in plants[29]. Interestingly, LRR-RLKs are also

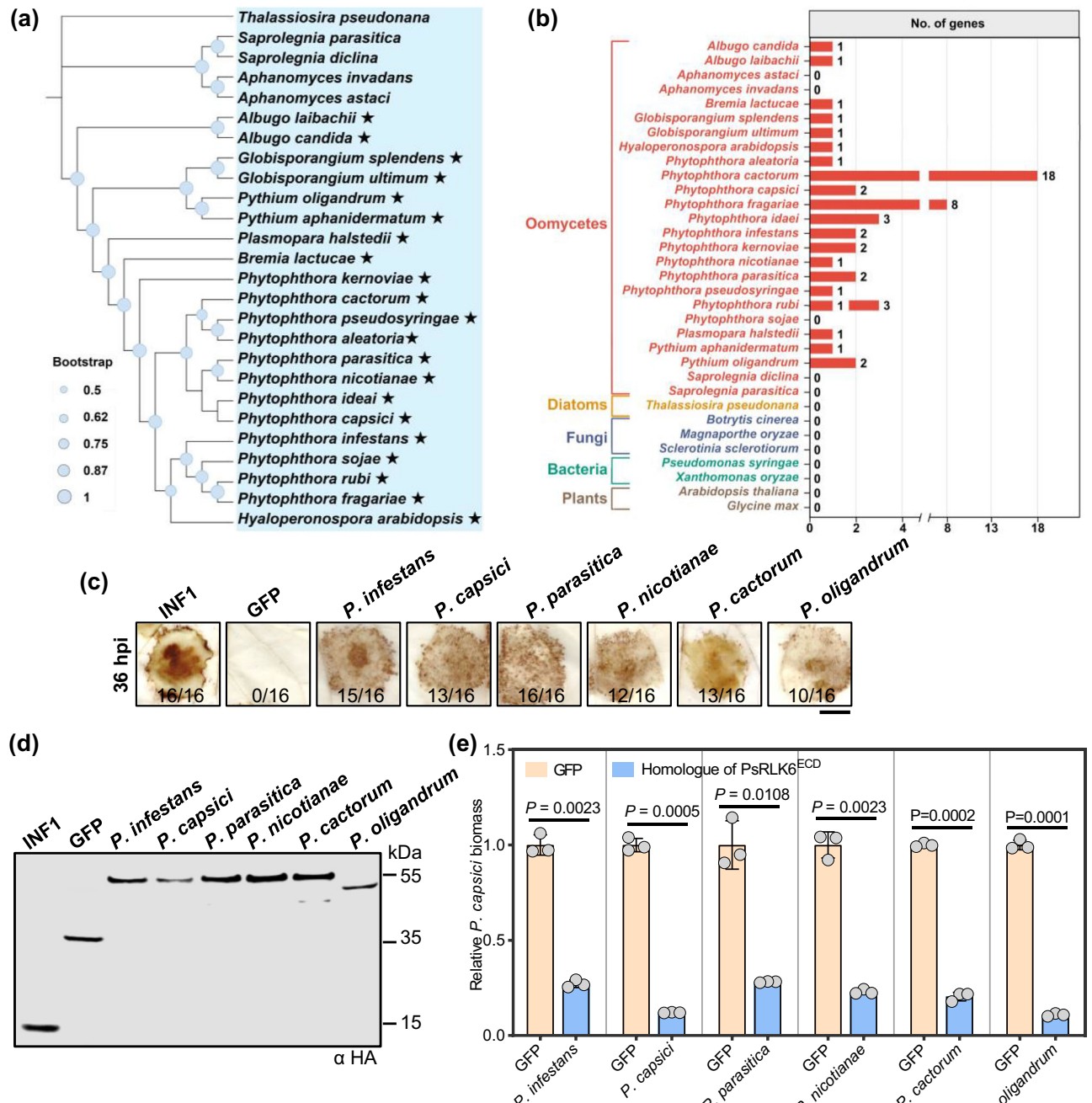

**Fig. 5 | Conserved PsRLK6 homologs that exhibit elicitor function in *N. benthamiana* are widely distributed in oomycetes. a** The phylogeny of selected species. Bootstrap percentage support for each branch is indicated. Maximum-likelihood phylogenetic tree of oomycete which contain the PsRLK6 homologs is shown. Asterisks represent the PsRLK6^ECD homologous sequences are present. **b** Distribution of PsRLK6^ECD homologous proteins. The number of homologous sequences in selected species are indicated. **c** ROS accumulation assays for transient expression of selected PsRLK6^ECD homologous genes from six oomycetes in *N. benthamiana* by agroinfiltration. INF1 and GFP were used as positive and negative

control, respectively (*n* = 16 biologically independent samples from 3 different biologically independent experiments). Ratios indicate the proportion of infiltrated sites that developed the ROS accumulation phenotype. Scale bar, 1 cm. **d** Immunoblotting showing the expression of corresponding proteins. **e** Transient expression of selected PsRLK6^ECD homologous genes in *N. benthamiana* enhanced resistance to *P. capsici* (*n* = 3 biologically independent experiments). Quantification of *P. capsici* infection by qRT-PCR analysis to measure the ratios of *P. capsici* to *N. benthamiana* DNA. Data are shown as mean ± SD. *P* values were derived by two-tailed Student's *t* test. Source data are provided as a Source Data file.

widespread in oomycetes and play important roles in development and stress[15–17]. Remarkably, here we found a *P. sojae* LRR-RLK (PsRLK6) and its homologs could be detected by plants to activate pattern-triggered immunity.

Sexual reproduction is a key process in the infection cycle of oomycetes, which produce oospores that are important inoculum to trigger a new epidemic in the next growing season[30]. However, the molecular mechanism of sexual reproduction in oomycetes is poorly

understood. Only very few genes have been reported to regulate the oospore development of oomycetes. In *Pythium ultimum*, knockout mutants of *PuM90*, a Puf family RNA-binding protein, showed significantly defective in oospore formation with empty oogonia or oospores larger in size with thinner oospore walls[31]. Further study indicated that a tripartite recognition motif in the Puf domain of PuM90 could specifically bind to the 3'-untranslated region region of PuFLP, which encodes a flavodoxin-like protein and thereby repress

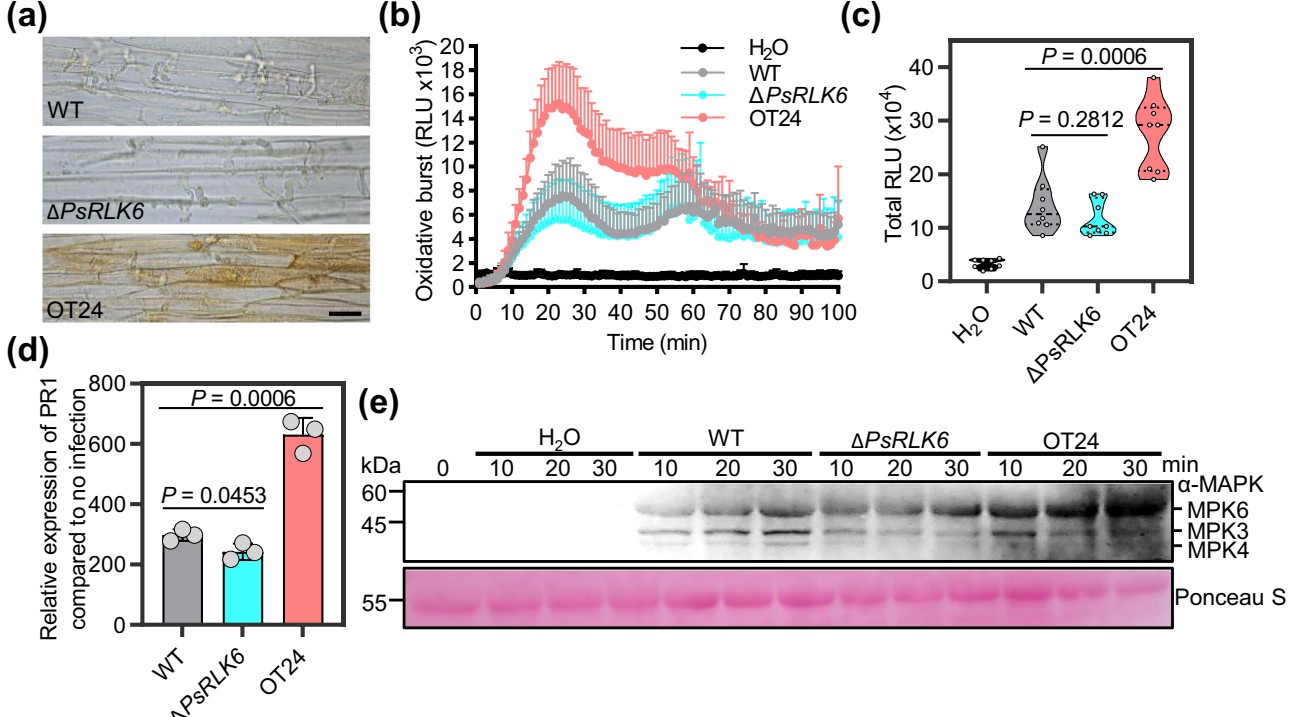

**Fig. 6 | PsRLK6 could induce PTI immune responses during infection.**
**a** Microscopic observations of invasive hyphae from different strains in epidermis of soybean hypocotyls at 24 hpi. DAB staining was performed on the epidermis of seedling hypocotyls. Bar, 20 μm. The experiment was repeated three times (more than 30 epidermal slices) with similar results. **b** ROS production in soybean leaf discs treated with *P. sojae* zoospores of indicated strains (10,000 zoospores per/ mL). Long-term ROS production was monitored for 100 min at 1-min intervals. Bars indicate mean ± SEM (*n* = 8 biologically independent samples). The experiment repeated three times with similar results. **c** Total ROS production during 0–40 min of continuous measurement. The data are shown as violin plots with individual data points plotted (*n* = 8 biologically independent samples). The center line, edges and whiskers indicate the median, lower and upper quartiles and the minimum and maximum, respectively. **d** The relative expression of *PR1a* at 12 h after inoculation of etiolated soybean seedlings by *P. sojae* wild-type, PsRLK6-knockout and overexpression lines compared to 0 h. Data are shown as mean ± SD (*n* = 3 biologically independent experiments). **e** MAPK activation of soybean leaves after treatment with indicated zoospores suspension was analyzed by immunoblot with the α-pTEpY antibody. Protein loading is shown using Ponceau S. The H₂O treatment was used as the negative control. The experiments were repeated three times with similar results. *P* values were derived by two-tailed Student's *t* test. Source data are provided as a Source Data file.

PuFLP mRNA level to facilitate oospore formation[31]. In *P. infestans*, silencing of a gene that encodes loricrin-like protein blocked oospore wall formation[32]. As for *P. sojae*, YPK1 (a serine/threonine protein kinase-encoding gene)[33], YKT6 (a soluble *N*-ethylmaleimide-sensitive factor attachment protein receptors)[34], GK5 (a G-protein-coupled receptor with a phosphatidylinositol phosphate kinase domain)[35], IMPA1 (a conserved importin *α*)[36], and *N*-glycosylation in HSP70[37] also play important roles in oospore production while the mechanisms of regulating oospore development remain largely elusive. Here, we showed that PsRLK6 plays an indispensable role in oospore production and morphological development. Knockout of *PsRLK6* led to reduced oospore number, increased ratio of abnormal oospores. During infecting soybean, Δ*PsRLK6* produced much less mature oospores compared to WT. Although knockout of *PsRLK6* does not affect the zoospore development and virulence[16], impaired oospore production and survival rate in soybean residues likely could not guarantee sufficient inoculum to trigger a new epidemic in the next growing season in the field.

Microbe-associated molecular patterns (MAMPs) are highly conserved molecules and have an essential function in microbial fitness or survival within the microbiome[3]. These molecules are usually involved in the growth, development, or pathogenicity of microbes. Classical MAMPs are usually structural molecules, such as bacterial flagellin, peptidoglycan, lipopolysaccharides, oomycete glucans, and fungal chitin[3]. Many virulence secretory proteins are also MAMPs, such as NLP[38], XEG1[39] PcEXLX1[40]. The reproduction of fungi and oomycetes is vital for the infection cycle and virulence variation, but only a few reports indicate that molecules involved in this process were identified as MAMPs. Zoospore of *Phytophthora* and conidiospore of fungi could directly elicit plant defense during infection[41,42]. Meanwhile, the MAMPs VmE02[43] and VdNLP1[38] regulate the conidiation of fungal pathogens *Valsa mali* and *Verticillium dahliae*, respectively. In this study, we first reported a protein that regulates the sexual reproduction of *Phytophthora* was recognized by plants. And we reveal that the transmembrane protein of pathogens also could be recognized as a MAMP.

The formation of heteromeric PRR complexes is important to activate intracellular signaling for immunity[44]. LRR-RLKs recruit the co-receptor BAK1 upon ligand binding, while LRR-RLPs require another co-receptor SOBIR1 besides BAK1[45]. PsRLK6-triggered immune responses depend on both BAK1 and SOBIR1 in *N. benthamiana*, suggesting that PsRLK6 is likely recognized by a plant LRR-RLP. In animals, transmembrane LRR proteins regulate synapse formation and function in the vertebrate nervous system by directly interacting with membrane proteins of neighboring cells[46]. During infection, pathogens release cell-wall degrading enzymes to enter the host's apoplast, and host plants also secret enzymes to degrade the cell wall of pathogens to limit infection[47,48]. In pathogen-host interactions, LRR-containing proteins could also mediate the infection cycle. For example, *Listeria monocytogenes*, a bacterial pathogen that causes gastroenteritis, abortions, and meningitis, produces about 25 different LRR-containing internalin proteins[49]. Biochemical and structural analyses

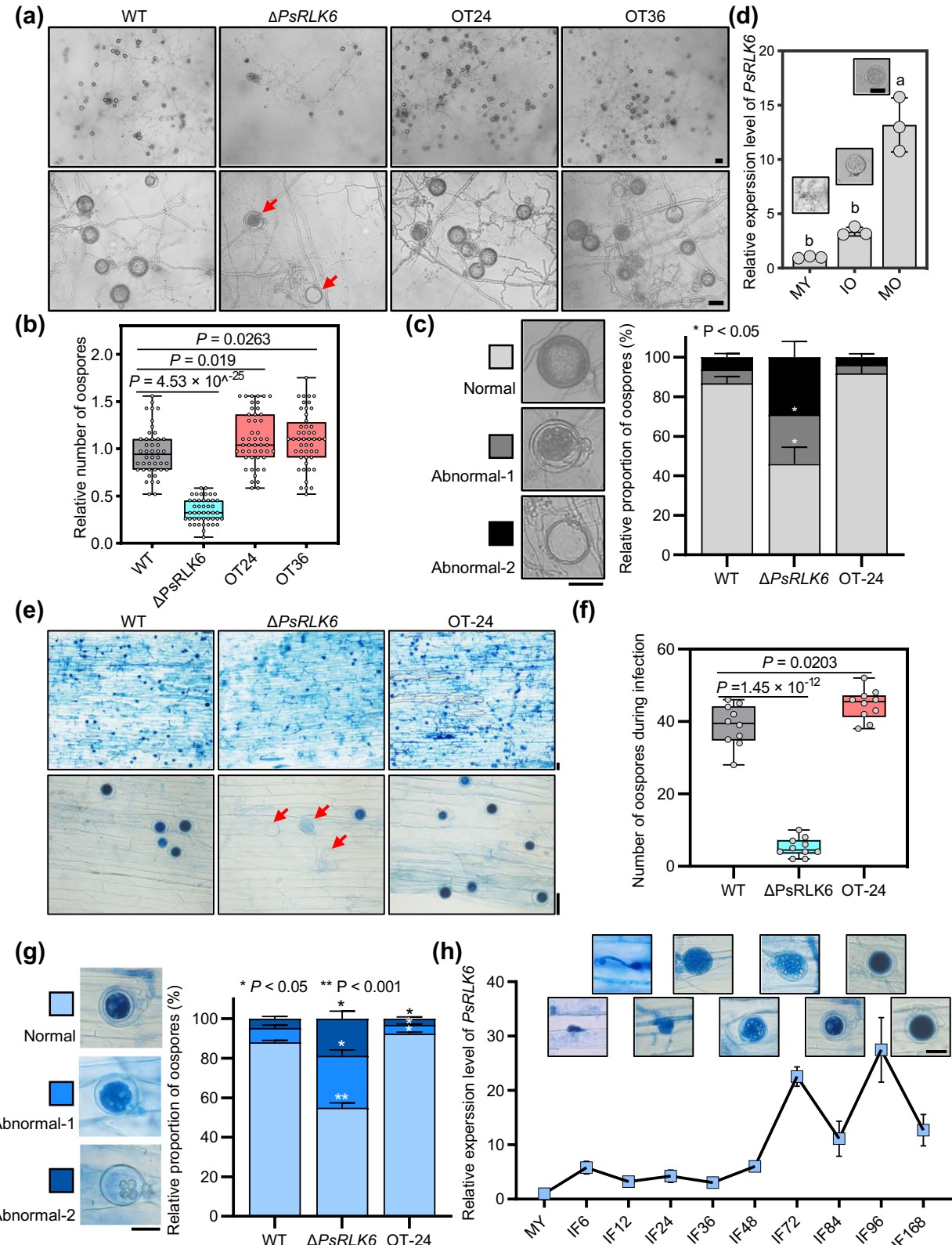

demonstrated that the LRR domains of cell-surface located internalin proteins bind directly to distinct human receptors to promote the internalization of bacteria into mammalian cells[49]. Thus, in the plant apoplast battleground, *P. sojae* LRR-RLKs might interact with plant receptors to mediate cell-to-cell communications.

Another possible scenario is that the ECD of PsRLK6 is cleaved by the plant and then released into the apoplast. In animals, controlled proteolysis is a well-known general mechanism that regulates the function of transmembrane receptors. Proteolytic cleavage of the extracellular portion of transmembrane proteins at or near the cell surface is referred as ectodomain shedding. It can be either constitutive or stimulus-induced and has been demonstrated for a lot of animal proteins, including cell adhesion molecules, growth factors, and receptors[50,51]. Prominent examples are receptor tyrosine kinases

**Fig. 7 | PsRLK6 regulates the oospore development of *P. sojae*. a** Morphology of oogonia and oospores generated by WT, *PsRLK6*-knockout (Δ*PsRLK6*) and *PsRLK6*-*GFP* overexpression transformants (OT-24/36). Oospores were generated on LBA medium for 14 days and observed under a light microscope. The arrows indicate the abnormal oospores. The experiment was repeated three times with similar results. **b** Statistical analysis of oospore number from 14-day-old cultures on LBA medium (*n* = 44–48 biologically independent samples). **c** The proportion of oospores with three different morphologies. **d** Transcript levels of *PsRLK6* measured using qRT-PCR in mycelia, IO (immature oospores) and MO (mature oospores) stages. Data are shown as mean ± SD (*n* = 3 biologically independent experiments). **e** Oospores were generated on infected root tissue at 48 hpi (top row) and 96 hpi (bottom row) stained with lactophenol-trypan blue. The arrows indicate the abnormal oospores. **f** Statistical analysis of oospore numbers from infected root

tissue at 48 hpi. Data are shown as mean ± SD (*n* = 10 biologically independent samples). **g** The proportion of oospores with three different morphologies during infection. Data are shown as mean ± SD (*n* = 3 biologically independent experiments). **h** Transcript levels of *PsRLK6* measured using qRT-PCR when mycelia were inoculated in etiolated soybean seedlings for 0, 6, 12, 24, 48, 72, 84, 96, and 168 h. The corresponding oospore morphology was shown above. Data are shown as mean ± SD. The experiment was performed three times with similar results. The statistical analyses were performed with two-tailed Student's *t* test. Different letters represent significant difference. *P < 0.05; **P < 0.01. In box plots in (**b**) and (**f**), the center line, box edges and whiskers indicate the median, lower and upper quartiles and the minimum and maximum, respectively. Scale bars, 20 μm. Source data are provided as a Source Data file.

epidermal growth factor receptors (EGFRs), which are structurally similar to plant RLKs and their ligands. Both EGFRs and EGFR ligands are synthesized as membrane-anchored precursors and their soluble ectodomains are released by proteolysis[52]. Moreover, numerous MAMPs need to be processed by plants before being recognized, such as bacterial flagellin processed by BGAL into flg22, fungal chitin and pectin are degraded into oligomers, and the oomycete-derived ceramide cleaved by plant apoplastic ceramidase into sphingoid base[5,53]. Taken together, we speculate that PsRLK6$^{ECD}$ might be released into the host's apoplast to be detected by the plant immune receptor.

Plant RLKs and RLPs form dynamic complexes in a ligand-dependent manner, which is essential for signaling activation[54,55]. The *Arabidopsis* LRR-RLKs show a complicated interaction network revealed by a high-throughput screen[19]. Likewise, we previously showed that *P. sojae* LRR-RLKs could form various complexes by split-luciferase assay[16]. Indeed, cell surface transmembrane proteins of animals interact directly to mediate cell-to-cell communications[46]. Here, we reveal that a *P. sojae* LRR-RLK is likely sensed by a plant LRR-RLP that forms a complex with LRR-RLKs BAK1 and SOBIR1 to activate plant immunity, suggesting that the pathogen receptor could also form complexes with plant receptors. The parasitic weed *Striga gesnerioides* produces an LRR effector to promote host colonization[56]. Therefore, it's interesting to further determine other *P. sojae* LRR-RLKs that could interact with plant receptors to disturb plant immunity.

The primary way to control plant diseases is using chemical fungicides, however, which threaten the environment and food safety. Microbe-associated molecular patterns (MAMPs) are environmentally friendly biological factors that enhance plant resistance to pathogens[57]. Here, we show that PsRLK6 induces immune responses in soybean and tomato, and the elicitor activity is heat-stable. Thus, PsRLK6 is a promising bio-stimulant for the control of plant diseases. Moreover, immune receptors that recognize MAMPs can be used for plant-resistance breeding. For instance, interfamily transfer of the LRR-RLK EFR that recognizes bacterial EF-Tu proteins in tomato and rice confers broad-spectrum resistance to bacteria[57]. Since RLK6 homologs that have elicitor activity are widespread in oomycete pathogens, further characterization of the corresponding receptor of RLK6 will provide resistance to these pathogens.

In summary, we report that an LRR-RLK from the oomycete pathogen *P. sojae* is essential for sexual reproduction by regulating oospore development. PsRLK6 triggers BAK1- and SOBIR1-dependent immune responses, representing a novel type of MAMP that shows the cell communication between cell surface receptors from plant and pathogen.

## Methods

### Microbial cultures

The *Phytophthora sojae* strains (P6497 and transformants), *P. capsici*, *P. parasitica*, *P. nicotianae*, *P. cactorum*, and *Pythium oligandrum* were routinely grown on 10% vegetable (V8) juice agar medium at 25 °C in the dark. The *P. infestans* strain T30-4 was routinely cultured on the

solid RSA/V8 medium at 18 °C in the dark. *P. sojae* mycelia and zoospores were prepared as previously described[16]. *PsRLK6* overexpression transformants were generated using the PEG-mediated protoplast transformation[26]. Overexpression lines were confirmed by immunoblot.

### Plant growth conditions and infection assays

In this study, *Nicotiana benthamiana* plants were grown in soil at 23 °C with a 16 h light and 8 h dark photoperiod. Soybean plants (Hefeng 47) were grown in soil at 23 °C with a 16 h light and 8 h dark photoperiod. Tomato seeds were germinated on a Petri dish lined with moist filter paper at 25 °C. After 5 days, the germinating sprouting seeds were transferred to soil and grown at 25 °C under a 16 h light and 8 h dark photoperiod. *Arabidopsis* plants were grown at 23 °C with a 10 h light and 14 h dark photoperiod.

For the *P. capsici* infection assay in *N. benthamiana*, leaves with different treatments were collected. *A*gar plugs of 5 mm diameter excised from fully grown mycelium of *P. capsici* were placed on adaxial surface of the leaves and incubated in darkness at 25 °C. The inoculated leaves were kept in transparent plastic boxes with high humidity and placed in a climate chamber. Lesion areas were measured at 36 h post-inoculation (hpi).

For the *P. capsici* infection assay in tomato. Tomato plants were sprayed with GFP and PsRLK6$^{ECD}$ proteins. After 12 h, leaves were excised. *A*gar plugs of 5 mm diameter excised from fully grown mycelium of *P. capsici* were placed on the adaxial surface of the leaves and incubated in darkness at 25 °C. After three days post inoculation, disease severity and *in planta P. capsici* biomass was quantified.

For the *P. capsici* infection assay in *Arabidopsis*. *Arabidopsis* Col-0 plants were sprayed with GFP and PsRLK6$^{ECD}$ protein. After 12 h, leaves were excised. About 150 zoospores of *P. capsici* were inoculated onto the center of a detached leaves and incubated in a growth room in dark at 25 °C. Inoculated leaves were photographed under UV light and the lesion areas were measured at 24 hpi. Relative quantification of *P. capsici* biomass in infected *Arabidopsis* leaves was performed to evaluate disease development.

For the *P. sojae* infection assay, ~200 zoospores of each transformant and the wild-type strain were inoculated onto the hypocotyls of etiolated soybean seedlings (susceptible Hefeng47 cultivar). Disease symptoms were scored at 48 h post-inoculation (hpi). At least three replications were performed for infected seedlings of each transformant. All assays were repeated at least three times.

### *P. sojae* growth, oospore production, and staining assays

To determine the growth rate, all tested strains were cultured on 10% V8 agar medium at 25 °C in the dark. Colony diameters were measured and photographed after seven days. To quantify oospore production, all tested strains were cultured on LBA agar medium in 7 cm petri plates at 25 °C in the dark. The oospores produced by different strains were photographed and counted with at least 30 images per strain.

The different morphology of oospores was photographed, and the content ratio was calculated.

Oospores of the homothallic species *P. sojae* were collected and separated by following the techniques as described previously[58]. Briefly, a polytron was used to homogenize the mating tissues collected from five dishes. The tissues were washed by 50 ml of sterile deionized water using five intervals of 2 min each at 4 °C. Then, oospores and hyphae were concentrated by centrifugation for 10 min at 4000 × *g*, and washed three times in water by spinning at 650 × *g* for 5 min. To assess the colonization of soybean tissues by oospores, infected epidermal cells were collected at 48 and 96 hpi and soaked in lactophenol-trypan blue [(10 mL lactic acid, 10 mL glycerol, 10 g phenol, and 10 mg trypan blue (purchased from Beijing Solarbio Science & Technology Co., Ltd.), dissolved in 10 mL distilled water], and then the infected epidermal cells were examined under an inverted microscope.

To examine colonization of soybean tissues by oospores of different strains, infected epidermal cells were collected at 6, 12, 24, 48, 72, 84, 96, 168 hpi and then soaked in lactophenol-trypan blue to stain for 2 h. After destaining in chloral hydrate and water, cells were examined under a light microscope. The morphology of oospores was observed and the number was measured. Each strain was tested using at least two different preparations of mycelia and five plants for infection. The lesion areas of plants infected by pathogen are measured by ImageJ v1.52 and the figures are made by GraphPad Prism 9 software.

## Bioinformatics

The proteomes of different organisms used in this study were obtained from National Center for Biotechnology Information (NCBI) databases with following accession numbers: *Phytophthora sojae* (GCA_000149755.2), *P. infestans* (GCA_000142945.1), *P. nicotianae* (GCA_001483015.1), *P. aleatoria* (GCA_018873745.1), *P. pseudosyringae* (GCA_019155715.1), *P. fragariae* (GCA_009733025.1), *P. rubi* (GCA_009732945.1), *P. capsici* (GCA_000325885.1), *P. idaei* (GCA_016880175.1), *P. cactorum* (GCA_003287315.1), *P. kernoviae* (GCA_001712705.2), *P. parasitica* (GCA_000509465.1), *Hyaloperonospora arabidopsis* (GCA_000173235.2), *Bremia lactucae* (GCA_004359215.2), *Plasmopara halstedii* (GCA_900000015.1), *Pythium ultimum* (GCA_000143045.1), *Pythium oligandrum* (GCA_005966545.1), *Pythium aphanidermatum* (GCA_000387445.2), *Globisporangium splendens* (GCA_006386115.1), *Albugo laibachii* (GCA_902706625.1), *Albugo candida* (GCA_001078535.1), *Aphanomyces invadans* (GCA_000520115.1), *Aphanomyces astaci* (GCA_003546625.1), *Saprolegnia diclina* (GCA_000281045.1), *Saprolegnia parasitica* (GCA_000151545.2), *Thalassiosira pseudonana* (GCA_000149405.2), *Botrytis cinerea* (GCA_000143535.4), *Magnaporthe oryzae* (GCA_000002495.2), *Sclerotinia sclerotiorum* (GCA_001857865.1), *Pseudomonas syringae* (GCA_002905815.2), *Xanthomonas oryzae* (GCA_008370835.2), *Arabidopsis thaliana* (GCA_001651475.1), and Glycine max (GCA_000004515.5). To determine the phylogenetic relationships of selected organisms, BLASTP (v2.5.0) comparisons (*E* value < 1e-5) of all the protein sequences derived from above organisms were used as input into TribeMCL v14.137[59] using an inflation value of 2.0 for clustering. Protein sequence clusters with only one member from each organism were defined as single-copy core proteins (Fig. 5a). The multigene phylogenetic tree was conducted by MEGA 11[60] following the maximum-likelihood based method with 1,000 bootstrap replicates based on the concatenated sequences of single-copy core proteins. Single copy genes used in the phylogenetic analysis was as listed in Supplementary Data 2. To search for PsRLK6 orthologs in other organisms, PsRLK6 was used as the query to search against proteomes above by reciprocal BLASTP with an *E*-value cut-off of 1e-5. A phylogenetic tree of PsRLK6 and its homologous proteins was constructed by

following the neighbor-joining algorithm with 1,000 bootstrap replicates using MEGA 11 software to identify PsRLK6 orthologs (Fig. 5b). The multiple sequence alignment analysis was performed using MUSCLE (v3.8.31) tool with default parameters. To analyze the amino acid conservation, we used the Consurf server (https://consurf.tau.ac.il/consurf_index.php) with the default parameters except uploading a multiple sequence alignment of obtained 53 homologous protein sequences[24] (Supplementary Figs. 5a and b). The structures were visualized in the PyMOL v2.4.0 program.

Protein structure prediction. The structure of PsRLK6$^{ECD}$ is predicted by ColabFold[23] v1.5.2: AlphaFold2 using MMseqs2 and the detect templates in pdb70 (Fig. 4a). All the ColabFold output was included with the Supplementary Data 3. The structures were visualized in the PyMOL v2.4.0 program.

## Plasmid construction

A mixture of cDNAs from *P. sojae* collected at different infection stages was used as a template to amplify the coding sequences of *P. sojae* genes using the Phanta Super-Fidelity DNA Polymerase (P501-d1, Vazyme). Genes were cloned into vectors based on homologous recombination technology using the Vazyme ClonExpress II One Step Cloning Kit (C112, Vazyme). For *Agrobacterium*-mediated transient expression in *N. benthamiana*, the coding sequences of *P. sojae LRR-RLKs*, *PsRLK6$^{ECD}$* homologs and truncated versions of *PsRLK6$^{ECD}$* were cloned into the vector pBIN-3xHA. pBIN:INF1 and pBIN:GFP plasmid constructs served as positive and negative controls, respectively. For expression of the protein in *Pichia pastoris*, *PsRLK6$^{ECD}$* and *PsRLK7$^{ECD}$* without the signal peptide were cloned into pPICZaA vector. For expression of the protein in *E. coli*, LRR5-6 region of PsRLK6$^{ECD}$ were cloned into pCold TF vector. To generate overexpression transformants, *PsRLK6* was cloned into the pTOR-GFP vector. The primers and constructs used in this study are listed in Supplementary Table 1.

## Transient expression in *N. benthamiana* and DAB staining

*Agrobacterium*-mediated transient expression assays were performed to test the ROS-inducing activity. The indicated constructs were transformed into *A. tumefaciens* strain GV3101. The *Agrobacterium* strains carrying a plasmid were cultured in the LB medium containing antibiotics at 28 °C for 16–18 h. And then cells were collected by centrifugation and resuspended in infiltration buffer (10 mM magnesium chloride (MgCl$_2$), 10 mM MES, 200 μM acetosyringone, pH 5.7) in the dark at 28 °C for 2 h. The *Agrobacterium* cell suspension was infiltrated into *N. benthamiana* leaves using a needleless syringe at a concentration of 0.5 OD$_{600}$. After *N. benthamiana* leaves were infiltrated with *Agrobacterium* for 24 h, the infiltrated leaves were soaked and stained with 3,3-diaminobenzidine (DAB, Sigma-Aldrich) solution for 8 h in the dark and then were destained with ethanol before observation.

## VIGS Assays in *N. benthamiana*

For VIGS assays, pTRV1, pTRV2:*BAK1*, or pTRV2:*GFP* plasmid constructs were introduced into *A. tumefaciens* GV3101. The cultured *Agrobacterium* cells were harvested and resuspended in infiltration buffer to an 0.5 OD$_{600}$. *Agrobacterium* strains harboring pTRV2:*BAK1* or pTRV2:*GFP* vector combining with that harboring pTRV1 vector were mixed in a 1:1 ratio. The cocultures were then infiltrated into two primary leaves of *N. benthamiana* at the four-leaf stage. The effectiveness of the VIGS assay was evaluated using the *PDS* gene as described[61]. The silencing efficiency of *BAK1* was validated using RT-qPCR analysis.

## Western blot analysis

For western blotting, protein extraction buffer (10 mM Tris/Cl [pH 7.5], 0.5 mM EDTA, 150 mM NaCl, 1% Triton X-100, 1% deoxycholate, 1% SDS) supplemented with protease inhibitor cocktail (Sigma) was used to extract the total proteins from plants. Anti-HA antibody (1:5,000;

Sigma) was used for immunoblotting protein with an HA tag. Anti-GFP antibody (1:5000; Abmart) was used for immunoblotting protein with an GFP tag.

## Expression and purification of recombinant proteins

The expression host *Pichia pastoris* KM71H (Muts) (Invitrogen) was cultured on yeast extract-peptone-dextrose medium. BMGY (buffered glycerol-complex medium) at pH 6.5 was used for yeast growth and BMMY (buffered methanol-complex medium) at pH 6.5 was used for induction (Easy Select Pichia expression kit; Invitrogen). *P. pastoris* transformants were screened for protein induction in 24-well plates as described[62]. Induction of protein expression was performed according to the manufacturer's instructions. Purification of recombinant GFP, PsRLK6$^{ECD}$, and PsRLK7$^{ECD}$ protein from the culture supernatant was performed by affinity chromatography using Ni-NTA Superflow resin. LRR5-6 region were recombinant expressed using *E. coli* strain BL21 (DE3) followed by induction with isopropyl β-d-1-thiogalactopyranoside (IPTG, 0.5 mM; Sangon Biotech, A600168) at 18 °C for 16 h. Then purification of recombinant LRR5-6 TF protein from the culture supernatant was performed by affinity chromatography using Ni-NTA Superflow resin. Subsequently, HRV 3 C Protease (Takara, 7360) was employed to cleave the labeled LRR5-6 protein and produce unlabeled LRR5-6 protein.

## Measurement of reactive oxygen species

Reactive oxygen species (ROS) production was monitored with an L-012/peroxidase-based assay on leaf discs collected from *N. benthamiana*, soybean, tomato, and *Arabidopsis* plants. The leaf discs were floated overnight on 200 μL of ddH$_2$O in a 96-well plate. The ddH$_2$O was replaced with a working solution [20 μM L-012 (Waco), 20 μg/mL peroxidase (Sigma-Aldrich), 1 μM flg22 (Sangon), or 1 μM purified protein reaction solution]. After the addition of the working solution, the plate was immediately moved to a GLOMAX96 microplate luminometer (Promega, Madison, WI, USA) for measurement of luminescence.

ROS detection in soybean leaves in response to zoospores of different *P. sojae* strains was monitored with an L-012 /peroxidase-based assays as previous reported with some minor modifications[63]. Briefly, Leaf discs was collected from 2-week-old soybean plants and then floated overnight on 200 μL of ddH$_2$O in a 96-well plate. The ddH$_2$O was replaced with a working solution [20 μM L-012 (Waco), 20 μg/mL peroxidase (Sigma-Aldrich), zoospores (10000 zoospores per mL) of WT, PsRLK6-knockout (ΔPsRLK6) or PsRLK6-GFP over-expression transformants (OT-24). After the addition of the working solution, the plate was immediately moved to a GLOMAX96 microplate luminometer (Promega, Madison, WI, USA) for measurement of luminescence.

## RNA extraction and quantitative PCR assays

Total RNA of all samples was extracted using a PureLink RNA mini kit. Approximately 900 ng RNA was used for reverse transcription with oligo (dT) primers. Then, the cDNA reaction mixture was diluted three times and 2 μL was used as the template in a 20 μL PCR reaction with SYBR qPCR Master Mix (Vazyme). qPCR was performed using an ABI Prism 7500 Fast Real-Time PCR system as per the manufacturer's instructions. The relative quantitative method ($2^{-\Delta\Delta Ct}$) was used to evaluate the quantitative variation.

## MAPK assays

Proteins were extracted from 5-week-old *N. benthamiana* leaves after treatment with purified protein. The leaves were then frozen in liquid nitrogen and stored at −80 °C. The frozen leaves were ground in liquid nitrogen and homogenized in protein extraction buffer (50 mM Tris−HCl, pH 7.5, 5 mM EDTA, 5 mM EGTA, 2 mM DTT, 10 mM sodium fluoride, 50 mM β-glycerolphosphate, 10% [v/v] glycerol,

complete proteinase inhibitor cocktail [Roche, Mannheim, Germany], and Phosstop phosphatase inhibitor cocktail (Roche). Phosphorylation of MAPK proteins was detected by immunoblotting with anti-phospho-p44/42 MAPK antibody (1:5,000, Cell Signaling Technology, Danvers, Massachusetts, USA; #9101) according to the manufacturer's protocol. Western blots were stained with Ponceau S to verify equal loading.

MAPK assays were conducted in soybean leaves in response to zoospores (10,000/mL) produced by different *P. sojae* strains. Leaf discs was collected from 2-week-old soybean plants and then floated overnight on 200 μL of ddH$_2$O in a 96-well plate. The ddH$_2$O was substitued with either fresh ddH$_2$O or zoospores (10,000 zoospores per mL) of WT, PsRLK6-knockout (ΔPsRLK6) or PsRLK6-GFP over-expression transformants (OT-24) and incubated for 10 min, 20 min, 30 min. Subsequently, protein extraction and MAPK detection were performed as described above.

## Reporting summary

Further information on research design is available in the Nature Portfolio Reporting Summary linked to this article.

## Data availability

All data are available within the article and supplementary Files. The proteomes of different organisms used in this study can be obtained from National Center for Biotechnology Information (NCBI) databases with following accession numbers: *Phytophthora sojae* (GCA_000149755.2) [https://www.ncbi.nlm.nih.gov/datasets/genome/GCF_000149755.1/], *P. infestans* (GCA_000142945.1), *P. nicotianae* (GCA_001483015.1), *P. pseudosyringae* (GCA_019155715.1), *P. fragariae* (GCA_009733025.1), *P. rubi* (GCA_009732945.1), *P. capsici* (GCA_000325885.1), *P. idaei* (GCA_016880175.1), *P. cactorum* (GCA_003287315.1), *P. kernoviae* (GCA_001712705.2), *P. parasitica* (GCA_000509465.1), *Hyaloperonospora rabidopsis* (GCA_000173235.2), *Bremia lactucae* (GCA_004359215.2), *Plasmopara halstedii* (GCA_900000015.1), *Pythium ultimum* (GCA_000143045.1), *Pythium oligandrum* (GCA_005966545.1), *Pythium aphanidermatum* (GCA_000387445.2), *Globisporangium splendens* (GCA_006386115.1), *Albugo laibachii* (GCA_902706625.1), *Albugo candida* (GCA_001078535.1), *Aphanomyces invadans* (GCA_000520115.1), *Aphanomyces astaci* (GCA_003546625.1), *Saprolegnia diclina* (GCA_000281045.1), *Saprolegnia parasitica* (GCA_000151545.2), *Thalassiosira pseudonana* (GCA_000149405.2), *Botrytis cinerea* (GCA_000143535.4), *Magnaporthe oryzae* (GCA_000002495.2), *Sclerotinia sclerotiorum* (GCA_001857865.1), *Pseudomonas syringae* (GCA_002905815.2), *Xanthomonas oryzae* (GCA_008370835.2), *Arabidopsis thaliana* (GCA_001651475.1), and Glycine max (GCA_000004515.5) [https://www.ncbi.nlm.nih.gov/datasets/genome/GCF_000004515.6/]. Source data are provided with this paper.

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

## Acknowledgements
The work was supported by the Natural Science Foundation of Jiangsu Province (BK20221000 to Z.Y.), the National Natural Science Foundation of China (32230089 to D.D. and 32202251 to Z.Y.), the China Agriculture Research System (CARS-21 to D.D.), and the Jiangsu Funding Program for Excellent Postdoctoral Talent (2022ZB343 to Z.Y.).

## Author contributions
D.D. and Z.Y. conceived and designed the research. H.Z. and D.S. performed bioinformatics analyses. Y.P., P.J., J.S., S.Z., R.X., H.Q., and W.D. performed experiments. Y.P. and P.J. analyzed data. Y.P., Z.Y., and D.D. wrote the manuscript. All authors read and approved the final manuscript.

## Competing interests
The authors declare no competing interests.
