## [Peer Review File · Nature Communications]

Reviewers' Comments:

Reviewer #1:

Remarks to the Author:

Pei et al. report their findings on leucine-rich repeat receptor-like kinases in both pathogens and their plant hosts and their cell-to-cell/immune interactions. I was asked to comment on the bioinformatics executed in the manuscript. As I have no expertise in plant or plant-pathogen biology, I will only comment on the bioinformatics.

The authors have performed an extensive set of analyses. However, the smaller set of bioinformatics analyses performed, were difficult to follow as they were only sparsely described and critical information for reproducing them was missing.

Based on the given descriptions and some guess-work, the bioinformatics analyses were likely correctly performed and improving their description and adding supplementary information required for reproduction would improve the manuscript considerably.

Issues:

- The authors should list software versions used, parameters of the software executed and provide output of the software as supplementary material. The authors should also provide the scripts that were executed to make sure that results are reproducible.
- AlphaFold2 predictions should be provided together with their pLDDT coloring. The pLDDT would help the reader to judge the quality of the predicted structure shown in Fig 4a.
- I am confused by missing residues shown near the C-terminal end. I don't think that either ColabFold or AlphaFold2 predictions can result in missing residues. This issue doesn't seem to occur in Fig. S5. Why are there missing residues? It would be very helpful if all the ColabFold output was included with the supplementary information.
- I tried to reproduce the structure prediction and was able to do so with some guess-work. I couldn't find the accession for PsRLK6 from *Phytophthora sojae*. Please add the *P. sojae* PsRLK6 sequence and accession to Table S1. Using the NCBI blast webserver, I found UWU45069.1 as the most likely accession of PsRLK6. I was able to generate a highly confident prediction of the PsRLK6 and its ECD with ColabFold, that looks very similar to Fig 4a and S5. Please add a subsection to the methods section "Bioinformatics" describing the procedure to obtain the predicted structure and the prediction quality.
- The description in the methods section "Bioinformatics" is very sparse in details. The authors state that they obtained genomes from "NCBI and JGI databases". Please list the accessions. Next the authors talk about a protein-based analysis with blastp without stating where the proteomes come from. Next the authors use TribeMCL, which seems to be unsupported and deprecated software since a long time. Is there a particular reason for using TribeMCL?
- Please reference the respective figures and supplementary information within the methods section "Bioinformatics".

Reviewer #2:

Remarks to the Author:

The manuscript by Pei et al. identified PsRLK6, an LRR-RLK from the soybean root and stem rot pathogen *Phytophthora sojae*, as an elicitor which activates pattern-triggered immunity in soybean, tomato and *N. benthamiana* plants. Interestingly, this RLK and its immunity-inducing activity are conserved in oomycetes. Further analysis showed a small region (LRR5-6) in PsRLK6 extracellular domain is sufficient to activate BAK1- and SOBIR1-dependent immune responses, suggesting that its receptor in *N. benthamiana* is likely an LRR-RLP. In addition, PsRLK6 was shown to be required for the oospore development of *P. sojae*. These findings are interesting and of general interest to the plant-microbe interaction field.

1. Is the same region of PsRLK6 (LRR5-6) recognized in tomato and soybean to activate immune responses?
2. A collection of tobacco rattle virus-based constructs targeting LRR receptor-like genes in *N.*

benthamiana were previously generated and used to identify RXEG1, the receptor-like protein recognizing XEG1 from *Phytophthora sojae*. Have the authors tried to identify the receptor for PsRLK6 in *N. benthamiana* using the same set of silencing constructs?

3. For the PsRLK6ECD treatments, it is often unclear whether it was applied by spray or infiltration. Please specify the treatments in the figure legends.

Reviewer #3:

Remarks to the Author:

The manuscript by Pei and colleagues investigates the molecular function of the *Phytophthora sojae* receptor-like kinase PsRLK6. Most of the study is carried out in *Nicotiana benthamiana*, i.e. outside the biological context of PsRLK6. For example, it is not clear whether BAK1 and SOBIR1 homologues exist in *Phytophthora*. The idea that metazoan cells can interact directly, establish contacts and communicate may be due to the absence of a cell wall. Oomycete infection structures always contain a cell wall, including haustoria. As a result, the possibility that a direct interaction occurs between *N. benthamiana* and *P. sojae* remains to be tested, which the authors did not. This is especially worrying given that the authors can transform their microorganism and that they suspect a possible role of PsRLK6 in sensing sexual hormones. Overall, I find the conclusions drawn by the authors a bit too strong compared to the evidence they present.

The manuscript body and the figures contain a significant number of spelling mistakes and formatting issues. I have listed a few examples below.

- The trees in Fig 5a and 5b are too small and hard to read.
- line 321, "Saprolegniales" should be spelled "Saprolegniales".
- Fig 3b, "Tomoto" is "Tomato".
- Fig 5b, "Bacterium" should be the plural "Bacteria".
- line 332 and Fig 5e, "cactarum" is "cactorum".
- Fig 6c and 6g, "normol" should be "normal".

Responses to Reviewers

Reviewer #1 (Remarks to the Author):

Pei et al. report their findings on leucine-rich repeat receptor-like kinases in both pathogens and their plant hosts and their cell-to-cell/immune interactions. I was asked to comment on the bioinformatics executed in the manuscript. As I have no expertise in plant or plant-pathogen biology, I will only comment on the bioinformatics.

The authors have performed an extensive set of analyses. However, the smaller set of bioinformatics analyses performed, were difficult to follow as they were only sparsely described and critical information for reproducing them was missing.

Based on the given descriptions and some guess-work, the bioinformatics analyses were likely correctly performed and improving their description and adding supplementary information required for reproduction would improve the manuscript considerably.

Re: We appreciate your suggestions and comments for helping us to improve the MS. We agree that our MS only used small set of bioinformatics analyses, which are sparsely described, and critical information was missed. Here, we fix this by improving descriptions and adding supplementary information. The detail information could be seen in the revised MS or the below.

Issues:

- The authors should list software versions used, parameters of the software executed and provide output of the software as supplementary material. The authors should also provide the scripts that were executed to make sure that results are reproducible.

Re: In this study, only several published software was used and none of new scripts were related. As suggested, the software versions, executed parameters, and output used in this study were added in the section of "Methods".

- AlphaFold2 predictions should be provided together with their pLDDT coloring. The pLDDT would help the reader to judge the quality of the predicted structure shown in Fig 4a.

Re: We accepted this suggestion by providing the pLDDT together with predicted structure in the revised Fig.4a.

- I am confused by missing residues shown near the C-terminal end. I don't think that either ColabFold or AlphaFold2 predictions can result in missing residues. This issue doesn't seem to occur in Fig. S5. Why are there missing residues? It would be very helpful if all the ColabFold output was included with the supplementary information.

Re: We also noticed this problem, and here, re-analyzed the structure of PsRLK6ECD using ColabFold. Accordingly, Fig. 4a and Fig. S5 were updated, in which the problem of missing residues was solved. In addition, all the ColabFold output was included with the supplementary information in re-submitted MS.

- I tried to reproduce the structure prediction and was able to do so with some guess-work. I couldn't find the accession for PsRLK6 from *Phytophthora sojae*. Please add the *P. sojae* PsRLK6 sequence and accession to Table S1. Using the NCBI blast webserver, I found UWU45069.1 as the most likely accession of PsRLK6. I was able to generate a highly confident prediction of the PsRLK6 and its ECD with ColabFold, that looks very similar to Fig 4a and S5. Please add a subsection to the methods section "Bioinformatics" describing the procedure to obtain the predicted structure and the prediction quality.

Re: Indeed, UWU45069.1 is precisely PsRLK6. In this study, we just generated a prediction of PsRLK6ECD with ColabFold. As suggested, the PsRLK6 sequence and accession (UWU45069.1) are listed in the revised Table S1 "Index 54". The procedure to obtain the predicted structure and the prediction quality were added in the methods section " Bioinformatics ".

- The description in the methods section "Bioinformatics" is very sparse in details. The authors state that they obtained genomes from "NCBI and JGI databases". Please list the accessions. Next the authors talk about a protein-based analysis with blastp without stating where the proteomes come from. Next the authors use TribeMCL, which seems to be unsupported and deprecated software since a long time. Is there a particular reason for using TribeMCL?

Re: As suggested, we added a series of details in the revised methods section "Bioinformatics". The accessions of obtained proteomes from "NCBI and JGI databases" are listed in the revised methods section "Bioinformatics" and seen below. These proteomes were used for blastp searches. We revised the methods section accordingly.

In this study, PsRLK6 homologs were identified by blastp searches, and we just utilized TribeMCL here to identify single-copy core proteins to construct the species tree for selected organisms.

Phytophthora sojae (GCA_000149755.2), *P. infestans* (GCA_000142945.1), *P. nicotianae* (GCA_001483015.1), *P. aleatoria* (GCA_018873745.1), *P. pseudosyringae* (GCA_019155715.1), *P. fragariae* (GCA_009733025.1), *P. rubi* (GCA_009732945.1), *P. capsici* (GCA_000325885.1), *P. idaei* (GCA_016880175.1), *P. cactorum* (GCA_003287315.1), *P. kernoviae* (GCA_001712705.2), *P. parasitica* (GCA_000509465.1), *Hyaloperonospora arabidopsis* (GCA_000173235.2), *Bremia lactucae* (GCA_004359215.2), *Plasmopara halstedii* (GCA_900000015.1), *Pythium ultimum* (GCA_000143045.1), *Pythium oligandrum* (GCA_005966545.1), *Pythium aphanidermatum* (GCA_000387445.2), *Globisporangium splendens* (GCA_006386115.1), *Albugo laibachii* (GCA_902706625.1), *Albugo candida* (GCA_001078535.1), *Aphanomyces invadans* (GCA_000520115.1), *Aphanomyces astaci* (GCA_003546625.1), *Saprolegnia diclina* (GCA_000281045.1), *Saprolegnia parasitica* (GCA_000151545.2), *Thalassiosira pseudonana* (GCA_000149405.2), *Botrytis cinerea* (GCA_000143535.4), *Magnaporthe oryzae* (GCA_000002495.2), *Sclerotinia sclerotiorum* (GCA_001857865.1), *Pseudomonas syringae* (GCA_002905815.2), *Xanthomonas oryzae* (GCA_008370835.2), *Arabidopsis thaliana* (GCA_001651475.1), and *Glycine max* (GCA_000004515.5).

- Please reference the respective figures and supplementary information within the methods section "Bioinformatics".

Re: Accepted. We referenced the respective figures (Fig. 4a/5a/5b) and supplementary information (Figs. S5a and b) within the methods section "Bioinformatics".

Reviewer #2 (Remarks to the Author):

The manuscript by Pei et al. identified PsRLK6, an LRR-RLK from the soybean root and stem rot pathogen *Phytophthora sojae*, as an elicitor which activates pattern-triggered immunity in soybean, tomato and *N. benthamiana* plants. Interestingly, this RLK and its immunity-inducing activity are conserved in oomycetes. Further analysis showed a small region (LRR5-6) in PsRLK6 extracellular domain is sufficient to activate BAK1- and SOBIR1-dependent immune responses, suggesting that its receptor in *N. benthamiana* is likely an LRR-RLP. In addition, PsRLK6 was shown to be required for the oospore development of *P. sojae*. These findings are interesting and of general interest to the plant-microbe interaction field.

Re: We appreciate your positive and professional comments.

1. Is the same region of PsRLK6 (LRR5-6) recognized in tomato and soybean to activate immune responses?

Re: A good question. To address it, we expressed and purified the protein of PsRLK6 (LRR5-6) region using *E. coli* and tested its ability to induce ROS activity in different plants. The results showed that PsRLK6 (LRR5-6) could induce ROS burst in tomato and soybean, as well as *N. benthamiana*, and were added as Fig. 4d/e/f. The MS was revised accordingly.

2. A collection of tobacco rattle virus-based constructs targeting LRR receptor-like genes in *N. benthamiana* were previously generated and used to identify RXEG1, the receptor-like protein recognizing XEG1 from *Phytophthora sojae*. Have the authors tried to identify the receptor for PsRLK6 in *N. benthamiana* using the same set of silencing constructs?

Re: Yes, we indeed identified the candidate receptor for PsRLK6 in *N. benthamiana* and a suspected receptor was successfully obtained. However, since this study focused on identifying a novel MAMP in oomycetes, we decide not to include the results about receptors and plan to report it in the future.

3. For the PsRLK6ECD treatments, it is often unclear whether it was applied by spray or infiltration. Please specify the treatments in the figure legends.

Re: PsRLK6ECD treatments were applied by spray or infiltration in different assays, and we have specified the treatments in the figure legends. In general, for the soybean hypocotyl infection experiment, PsRLK6ECD treatment was applied by soak, for tomato and *Arabidopsis* infection experiments, PsRLK6ECD treatments was applied by spray. For all other experiments involving PsRLK6ECD treatments were all applied by infiltration.

Reviewer #3 (Remarks to the Author):

The manuscript by Pei and colleagues investigates the molecular function of the *Phytophthora sojae* receptor-like kinase PsRLK6. Most of the study is carried out in *Nicotiana benthamiana*, i.e. outside the biological context of PsRLK6.

Re: We agree that most of the study is carried out in *Nicotiana benthamiana*, which is a model plant widely used for MAMP identification. However, we also demonstrated that PsRLK6ECD treatment could trigger PTI immune responses in host soybean, including ROS burst, up-regulation of PTI marker gene *PR1* and induced resistance to *P. sojae* in soybean in previously MS. Especially, we detected the difference of host PTI induced activity including ROS burst, *PR1* expression and MAPK activation of different strains of *P. sojae* (WT, PsRLK6-knockout mutant Δ PsRLK6 and PsRLK6 overexpressing strain OT24). The new results showed that Δ PsRLK6 induces slightly weaker PTI activity compared to WT, while OT24 elicit significantly more PTI responses. We added a new Figure (Fig. 6) to present these findings and updated the MS accordingly.

For example, it is not clear whether BAK1 and SOBIR1 homologues exist in *Phytophthora*. **Resopnse: BAK1 and SOBIR1 are conserved co-receptors (LRR-RLK) in plants including *Nicotiana benthamiana* and soybean but not in *Phytophthora*. Indeed, *Phytophthora* and plant LRR-RLKs evolve independently (Si et al, *iScience*, 2021). Soybean has BAK1 and SOBIR1 homologues, which have been reported to regulate PTI as co-receptors (Wang et al, *EMBO Rep*, 2020).**

Reference

Si, J. et al. *Phytophthora sojae* leucine-rich repeat receptor-like kinases: diverse and essential roles in development and pathogenicity. *iScience*. 24,102725 (2021).

Wang, D. et al. A malectin-like receptor kinase regulates cell death and pattern-triggered immunity in soybean. *EMBO Rep*. 21, e50442 (2020).

The idea that metazoan cells can interact directly, establish contacts and communicate may be due to the absence of a cell wall. Oomycete infection structures always contain a cell wall, including haustoria. As a result, the possibility that a direct interaction occurs between *N. benthamiana* and *P. sojae* remains to be tested, which the authors did not.

Re: We really appreciate this constructive comment. The idea that transmembrane receptors between metazoan cells can interact and communicate prompted us to determine whether receptors of oomycetes and plants could also communicate. We agree that the cell wall is present in both plants and *phytophthora*, but cell wall composition is dynamically changing during interaction, and the pathogen need to make a trade-off between cell wall strength and stealth invasion due to cell wall components contain large amounts of PAMPs. Moreover, plant and pathogen produce large amounts of proteases during infection, which degrade the other's cell wall, leading to the possibility of direct contact between plant and pathogen. If the cell wall is always present, then large amounts secreted proteins cannot break through this barrier. However, due to the unidentified receptor in the current stage, it is difficult to

prove. In this MS, we just proposed a possible scenario where PsRLK6 could directly contact receptors in plants to trigger immunity.

Take a step back talk, we proposed another scenario for how PsRLK6 could trigger immune responses during infection in “Discussion. Based on the experimental data, we prefer to believe that the ECD of PsRLK6 is cleaved by the plant and then released into the apoplast. In fact, the PsRLK6 protein seemed to be cleaved when expressed in both *P. sojae* and *N. benthamiana* (see the figure below). However, further research is needed to identify exactly how PsRLK6 functions in this process. But either way, it doesn't affect our conclusion that PsRLK6 serves as a novel MAMP to trigger immunity in plants. To prevent misunderstanding, we also removed the words “contact directly to” “directly interact” throughout the MS.

This is especially worrying given that the authors can transform their microorganism and that they suspect a possible role of PsRLK6 in sensing sexual hormones.

Re: In this study, we found knockout of PsRLK6 led to reduced oospore number, increased ratio of abnormal oospores compared to WT. In previously MS, we suspected a possible role of PsRLK6 in sensing sexual hormones in “Discussion” to regulate sexual hormones due to PsRLK6 is a membrane receptor. We carefully considered about this discussion and decided to delete the discussion about “sensing sex hormones” in the light of the lack of evidence. The MS was revised according.

Overall, I find the conclusions drawn by the authors a bit too strong compared to the evidence they present.

Re: We agree with this suggestion by toning down our conclusions seen above.

The manuscript body and the figures contain a significant number of spelling mistakes and formatting issues. I have listed a few examples below.

- The trees in Fig 5a and 5b are too small and hard to read.- line 321, "Saprolegnlales" should be spelled "Saprolegniales".
- Fig 3b, "Tomoto" is "Tomato".
- Fig 5b, "Bacterium" should be the plural "Bacteria".
- line 332 and Fig 5e, "cactarum" is "cactorum".
- Fig 6c and 6g, "normol" should be "normal".

Re: All accepted. We proofread our manuscript carefully again and corrected the corresponding mistakes.

Reviewers' Comments:

Reviewer #1:

Remarks to the Author:

As my limited expertise within plants or their pathogens has not changed, I continue to only comment on the bioinformatics. From my side, the remaining issues are minor and can be resolved during the editing stage.

Minor issues remaining:

- I recommend using a maximum-likelihood based method instead of neighbor-joining to build the phylogenetic tree
- Please release the list of single copy genes used in the phylogenetic analysis
- Some software versions are still missing (MUSCLE, TribeMCL, blastp). I recommend including all software versions used in the reporting summary, given that several have already been listed there.
- I noticed that some of the species listed among the proteomes and in 5b have a different name in the NCBI taxonomy than the one listed (e.g. *Pythium ultimum* -> *Globisporangium ultimum*). As mentioned, I don't know much about the conventions of the field, but is this intended? I did not check all names.
- Small typo: Supplemental Date1 -> Supplemental Data 1

Additional comment that is likely out of scope for the current manuscript, but might be interesting for future studies:

- Since the author have now a high-quality predicted structure for the protein of interest, they might follow up on that by using structure based analysis with e.g. Foldseek. Using the Foldseek webserver (search.foldseek.com) I can find many low E-value remote hits in the AlphaFold database. Looking up the top hit (AF-A0A6A3X1E5-F1-model_v4) in the clustered AlphaFold database (<https://cluster.foldseek.com/cluster/A0A484DX50>) also reveals many from a cursory glance interesting looking homologs that seem to have the sheet-motif shown in Fig. 4a,S5a/b conserved.
- I recommend predicting full length sequences with AF2 or ColabFold. Additional information, generally leads to better structure predictions. For presentation, the region of interest can then be isolated afterwards.

Reviewer #2:

Remarks to the Author:

My suggestions have been adequately addressed.

Reviewer #3:

Remarks to the Author:

The authors revised their MS and toned down the claims the reported evidence did not support. However, the biological significance of the heterologous expression of a *Phytophthora* protein in plants is still unclear to me. I am still wondering why the authors did not perform their study in *Phytophthora sojae*, which is amenable to transformation and CRISPR/Cas gene editing.

Responses to Reviewers

Reviewer #1 (Remarks to the Author):

As my limited expertise within plants or their pathogens has not changed, I continue to only comment on the bioinformatics. From my side, the remaining issues are minor and can be resolved during the editing stage.

Re: Thank you for your constructive suggestions to improve the manuscript.

Minor issues remaining:

- I recommend using a maximum-likelihood based method instead of neighbor-joining to build the phylogenetic tree

Re: Based on your suggestion, the phylogenetic tree was rebuilt using a maximum-likelihood based method. We updated the graph in revised Fig. 5a.

- Please release the list of single copy genes used in the phylogenetic analysis

Re: We have added the list of single copy genes used in the phylogenetic analysis as Supplementary Data 3.

- Some software versions are still missing (MUSCLE, TribeMCL, blastp). I recommend including all software versions used in the reporting summary, given that several have already been listed there.

Re: The software versions of MUSCLE, TribeMCL, blastp have been provided in revised METHODS section. All software versions used in this study also provided in the reporting summary. We appreciate your construction advice.

- I noticed that some of the species listed among the proteomes and in 5b have a different name in the NCBI taxonomy than the one listed (e.g. *Pythium ultimum* -> *Globisporangium ultimum*). As mentioned, I don't know much about the conventions of the field, but is this intended? I did not check all names.

Re: In 2010, based on phylogeny and morphology, the genus *Pythium* (s. str.) is emended, and four new genera, *Ovatisporangium*, *Globisporangium*, *Elongisporangium*, and *Pilasporangium*, are described and segregated from *Pythium* s. lato (Uzuhashi et al, 2010). The latest name of *Pythium ultimum* is *Globisporangium ultimum*. *G. ultimum* = *P. ultimum*. We have revised this name in Figure 5a and b. Meanwhile, we check all names and just the name of *Pythium ultimum* has this situation.

Isolate no.	Isolate origin			Species ^a	GenBank accession No.		
	Origin	International	Substrate		Locality	ITS	D1/D2
UZ174	MAFF 241113	Soil (cultivated)	Kagoshima, Japan	G. splendens = P. splendens	AB468778	AB468716	AB468903
UZ307	MAFF 241114	Soil (uncultivated)	Kyoto, Japan	G. sylvaticum = P. sylvaticum G. ultimum = P. ultimum	AB468779	AB468717	AB468904

Ref: Shihomi Uzuhashi, Motoaki Tojo, Makoto Kakishima, Phylogeny of the genus *Pythium* and description of new genera, *Mycoscience*, 2010, <https://doi.org/10.1007/S10267-010-0046-7>.

- Small typo: Supplemental Date1 -> Supplemental Data 1

Re: Revised.

Additional comment that is likely out of scope for the current manuscript, but might be interesting for future studies:

- Since the author have now a high-quality predicted structure for the protein of interest, they might follow up on that by using structure based analysis with e.g. Foldseek. Using the Foldseek webserver (search.foldseek.com) I can find many low E-value remote hits in the AlphaFold database. Looking up the top hit (AF-A0A6A3X1E5-F1-model_v4) in the clustered AlphaFold database (<https://cluster.foldseek.com/cluster/A0A484DX50>) also reveals many from a cursory glance interesting looking homologs that seem to have the sheet-motif shown in Fig. 4a,S5a/b conserved.

Re: We appreciate your suggestions regarding structure-based analysis. It is worth noting that the LRR domain is present in a wide range of organisms, including animals, plants, fungi and oomycetes etc. Although the structural features are conserved across these domains, their functions vary significantly. In our future research endeavors, we will identify proteins that share similarity with PsRLK6ECD in plants based on structure-based analysis, which may play a role in plant immunity.

- I recommend predicting full length sequences with AF2 or ColabFold. Additional information, generally leads to better structure predictions. For presentation, the region of interest can then be isolated afterwards.

Re: We predicted full length sequences of PsRLK6 with ColabFold. However, PsRLK6 is a single transmembrane protein and the full-length protein's protein fold appears to be incorrect (as shown in the figure below). Then, we compared the extracellular protein structure predicted for the full-length protein with that predicted specifically for the extracellular region, and found a near-perfect overlap between their structures, so we just used the extracellular protein structure predicted by ColabFold.

Reviewer #2 (Remarks to the Author):

My suggestions have been adequately addressed.

Re: Thank you for your constructive suggestions to improve the manuscript.

Reviewer #3 (Remarks to the Author):

The authors revised their MS and toned down the claims the reported evidence did not support. However, the biological significance of the heterologous expression of a *Phytophthora* protein in plants is still unclear to me. I am still wondering why the authors did not perform their study in *Phytophthora sojae*, which is amenable to transformation and CRISPR/Cas gene editing.

Re: Thank you for your constructive suggestions to improve the manuscript. To address your confusion, we will answer the question from the following two points. Firstly, we generated *PsRLK6* knockout and overexpression mutants to investigate its roles in oospore production and PTI activation in soybean. However, since the gene editing efficiency of *Phytophthora* is very low, some heterologous expression assays were performed to test the plant immunity-inducing activity of *PsRLK6* homologs and truncated mutant variants, instead of CRISPR/Cas gene editing. Secondly, during *phytophthora* infection, a large amount of PAMPs are produced into plant apoplast and recognized by plant membrane receptors to trigger immunity. Thus, heterologous expression of a pathogen protein in plants is a commonly used method for studying whether the protein could trigger plant immunity.